# Multivalency regulates activity in an intrinsically disordered transcription factor

**Sarah Clark[1], Janette B Myers[2], Ashleigh King[3,4], Radovan Fiala[5], Jiri Novacek[5], Grant Pearce[6], Jörg Heierhorst[3,4], Steve L Reichow[2], Elisar J Barbar[1]\***

[1]Department of Biochemistry and Biophysics, Oregon State University, Oregon, United States; [2]Department of Chemistry, Portland State University, Oregon, United States; [3]St. Vincent's Institute of Medical Research, The University of Melbourne, Victoria, Australia; [4]Department of Medicine, St. Vincent's Health, The University of Melbourne, Victoria, Australia; [5]Central European Institute of Technology, Masaryk University, Brno, Czech Republic; [6]School of Biological Sciences, University of Canterbury, Christchurch, New Zealand

**Abstract** The transcription factor ASCIZ (ATMIN, ZNF822) has an unusually high number of recognition motifs for the product of its main target gene, the hub protein LC8 (DYNLL1). Using a combination of biophysical methods, structural analysis by NMR and electron microscopy, and cellular transcription assays, we developed a model that proposes a concerted role of intrinsic disorder and multiple LC8 binding events in regulating LC8 transcription. We demonstrate that the long intrinsically disordered C-terminal domain of ASCIZ binds LC8 to form a dynamic ensemble of complexes with a gradient of transcriptional activity that is inversely proportional to LC8 occupancy. The preference for low occupancy complexes at saturating LC8 concentrations with both human and *Drosophila* ASCIZ indicates that negative cooperativity is an important feature of ASCIZ-LC8 interactions. The prevalence of intrinsic disorder and multivalency among transcription factors suggests that formation of heterogeneous, dynamic complexes is a widespread mechanism for tuning transcriptional regulation.
DOI: https://doi.org/10.7554/eLife.36258.001

**\*For correspondence:**
barbare@oregonstate.edu

**Competing interests:** The authors declare that no competing interests exist.

## Introduction

Regulation of transcriptional activity is essential for every biological process. Common mechanisms for regulation include post-translational modifications and/or cooperativity among multiple activators and repressors (*Banerjee and Zhang, 2003*). Some transcription factors contain multiple regulatory sites for either post-translational modifications (*Meek and Anderson, 2009*) or binding partners (*Cantor and Orkin, 2002*), and their activity is thus tuned by the combined action of these components. Recent studies have revealed a high degree of intrinsic disorder in transcription factors, indicating that the inherent dynamical behavior harbored by these structures is critical for these regulatory events to take place (*Li et al., 2017*; *Liu et al., 2006*; *Minezaki et al., 2006*). Our developing understanding suggests that the intrinsically disordered domains in transcription factors may provide a multivalent platform for the recruitment of regulatory binding partners (*Currie et al., 2017*; *Shammas, 2017*). While the functional consequence of multivalent binding to an intrinsically disordered region has been described for a few transcription systems (*Dyson and Wright, 2016*; *Uversky et al., 2009*), it remains unclear for the vast majority of cases.

Human ASCIZ (ATMIN-Substrate Chk-Interacting Zn²⁺ finger) is an 88 kDa protein that has recently been identified as a transcription factor for the hub protein, LC8 (dynein light chain 8)

**eLife digest** Proteins help to regulate almost every process in the body, and come in various forms, sizes and purposes. Cells contain thousands of different proteins, but not every protein is needed at all times. To create new proteins, the information on a gene first needs to be transcribed into RNA (template molecules of the DNA) in a process known as transcription. A complex machinery inside the cell then uses the copy as a template to assemble the protein.

So-called transcription factors (also proteins) can switch the copying process on or off by binding to the start point of a gene. They can act alone or in complex with other proteins. The transcription factor called ASCIZ, for example, helps to regulate the production of a protein called LC8. LC8 attaches to more than 100 different proteins and plays an important role in many cell processes. Therefore, fine-tuning its production is essential.

The shape of a protein is critical to its purpose. Like most proteins, transcription factors are made up of chains of amino acids that fold into a specific three-dimensional (3D) structurewith a region that recognizes and binds to a specific DNA sequence. But many transcription factors also contain flexible, 'disordered' regions that do not fold into a rigid 3D shape. These may help to control the activity of genes, but their exact role is unclear.

ASCIZ contains an exceptionally long, disordered region that has multiple positions for binding LC8 along its chain. Previous research has shown that ASCIZ binds to the LC8 gene and increases transcription to produce more LC8 proteins. Once the protein levels are high enough, LC8 is thought to bind to the disordered region of ASCIZ and switch off transcription. Human ASCIZ proteins have 11 binding sites for LC8 molecules, while fruit flies have seven. Until now it was not clear why so many different binding sites exist.

To address this question, Clark et al. combined biophysical, structural and molecular biology techniques to analyze proteins from humans and fruit flies and to test their role in human cells. This revealed that LC8 and ASCIZ form a dynamic mixture of complexes, instead of a single fully-occupied complex. As the number of LC8 molecules bound to ASCIZ increased, the rate of transcription dropped. However, all of the binding sites were rarely fully occupied. Instead, three to four attached LC8 molecules seemed to be sufficient to ensure that LC8 levels remain balanced. When the number of LC8 molecules exceeded this value, the attachment rate for additional LC8 slowed down. So, even when there was an excess of LC8, most of the human ASCIZ binding sites were only partially filled. This way, the production of LC8 proteins was slowed, rather than fully shut down. As a result, the cells were able to fine-tune the transcription rate of LC8 and maintain a stable and balanced pool of these proteins.

This work suggests that disordered regions on transcription factors could help to keep cellular systems steady in the face of changing conditions. In the future, the combination of methods used here could reveal new information about other proteins with disordered regions.
DOI: https://doi.org/10.7554/eLife.36258.002

(*Jurado et al., 2012a*). Mice with mutations in ASCIZ that prevent LC8 transcription die in late embryogenesis and exhibit serious developmental defects in kidneys and lungs (*Goggolidou et al., 2014a*; *Goggolidou et al., 2014b*; *Jurado et al., 2010*). *Drosophila* ASCIZ knockouts die in early embryogenesis and localized knockdowns using RNAi show mitotic defects (*Zaytseva et al., 2014*). Mutant phenotypes in *Drosophila*, developing mouse B lymphocytes, and cultured cells are rescued by ectopic overexpression of LC8, demonstrating that the observed defects of ASCIZ knockouts are due to ASCIZ regulation of LC8 expression (*Goggolidou et al., 2014b*; *Jurado et al., 2012b*; *Zaytseva et al., 2014*). In addition, it has recently been shown that a conditional knockout of LC8 almost perfectly copies the corresponding phenotypes of ASCIZ knockouts in mouse B cell development and B cell lymphomagenesis (*King et al., 2017*; *Wong et al., 2016*).

LC8 is a highly conserved 20.6 kDa protein homodimer (10.3 kDa monomer) that facilitates self-association of its primarily disordered partners (*Barbar, 2008*; *Barbar and Nyarko, 2014*; *Clark et al., 2015*) (*Figure 1a*). LC8 binding is associated with a range of cellular processes, from cell division to apoptosis, underscoring LC8's essential role as a regulatory hub (*Dunsch et al., 2012*; *Puthalakath et al., 1999*). LC8 preferentially binds to a 10-amino acid motif in intrinsically

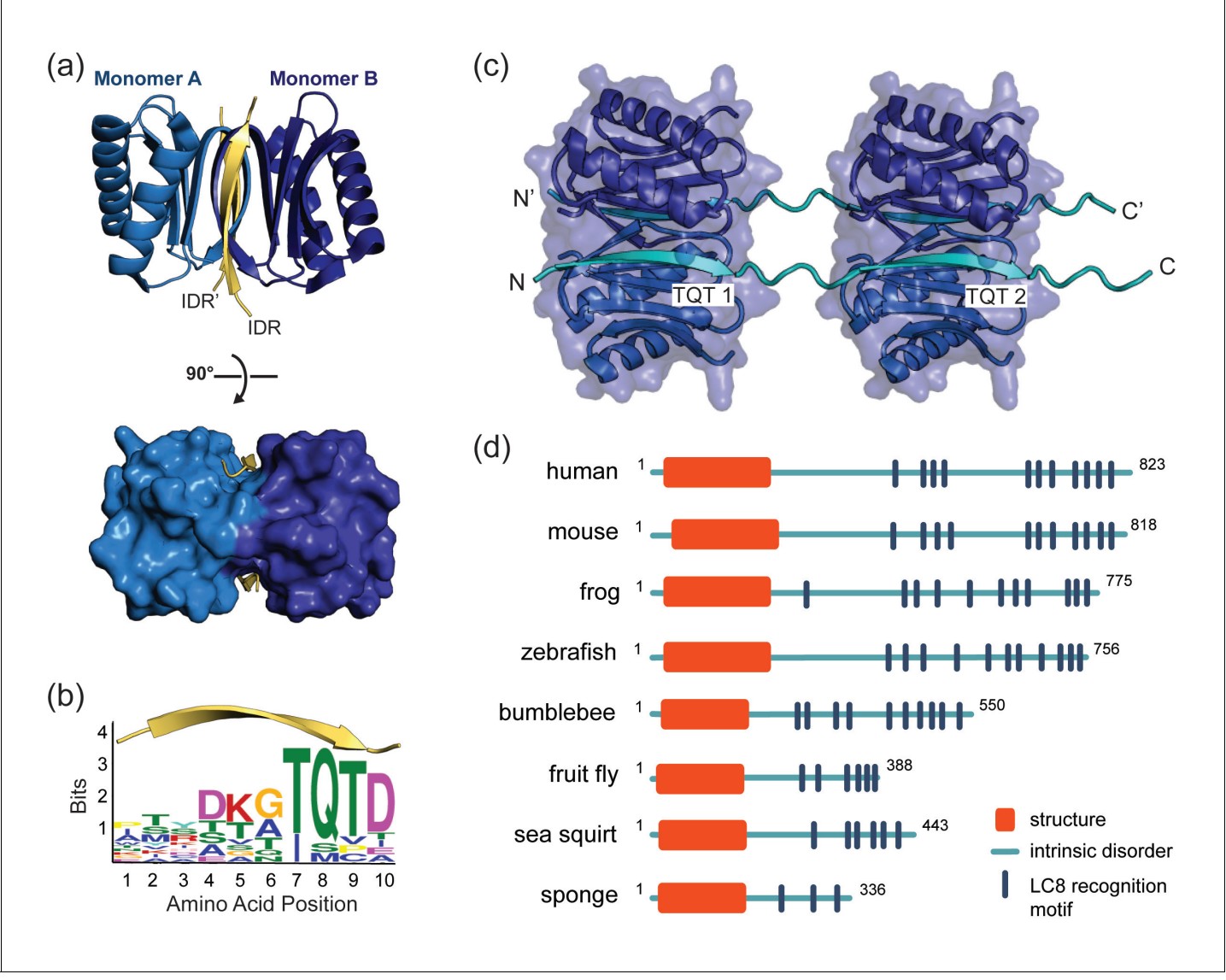

**Figure 1.** LC8 dimerizes its protein partners (**a**) Ribbon diagram of dimeric LC8, where each monomer is colored a different shade of blue and bound to a representative peptide in yellow (PDB 5E0L). A single LC8 dimer binds two peptides (intrinsically disordered region, IDR), one on each side of the dimerization interface, that are arranged in a parallel fashion. (**b**) A sequence logo of LC8 binding motifs derived from sequences of the motifs in the 11 crystal structures reported for LC8/peptide complexes. Height of amino acids indicates their relative frequency at that position. (**c**) A crystal structure of two LC8 dimers bound to two copies of the same intrinsically disordered chains illustrates how LC8 can bind to multivalent partner proteins with two LC8 recognition motifs, TQT1 and TQT2 (PDB 3GLW). (**d**) Sequence-based predictions of order (red boxes), disorder (blue lines), and LC8 binding motifs (dark blue bars) are shown for 10 ASCIZ proteins from the animal kingdom. ASCIZ proteins for different species were identified from a BLAST search (**Boratyn et al., 2013**) against the human protein. Sequence predictions of order and disorder were obtained with PSIPRED (**McGuffin et al., 2000**), where a criteria for order is based on >10% probability of predicted structure in a 50 + amino acid stretch. LC8 binding sites for human ASCIZ and *Drosophila* ASCIZ were obtained from the literature (**Rapali et al., 2011a**; **Zaytseva et al., 2014**). Putative LC8 binding sites for the other species were identified based on the presence of TQT residues.

DOI: https://doi.org/10.7554/eLife.36258.003

The following source data is available for figure 1:

**Source data 1.** A list of the 10-amino acid motifs and PDB identification codes for the 11 LC8-peptide crystal structures that were used to generate the sequence logo in *Figure 1b*.

DOI: https://doi.org/10.7554/eLife.36258.004

disordered regions (IDRs) containing highly conserved TQT residues at positions 7–9 (*Barbar, 2008*; *Rapali et al., 2011b*) (*Figure 1b*). In complex with LC8, the otherwise intrinsically disordered motif adopts a ß-strand conformation (*Benison et al., 2007*; *Liang et al., 1999*) (*Figure 1a,b*). Analysis of the 11 crystal structures of LC8 bound to short peptides containing the motif explains why the TQT residues are essential for binding: the Q is involved in interactions with both LC8 subunits in the dimer, while both T's are fully buried and thus evolutionarily constrained. In these interactions, TQT acts as the motif anchor while the other seven highly variable motif residues modulate affinity, as described in the *anchored flexibility model* of LC8 motif recognition (*Clark et al., 2016*). LC8 binds one motif in each of its two symmetrical binding grooves (*Figure 1a*), creating an IDP duplex that serves as a bivalent scaffold. This scaffold aids in higher order complex assembly by promoting binding of other proteins, including additional LC8 dimers (*Figure 1c*), and enhancing self-association and oligomerization processes that often involve coiled-coil formation (*Hall et al., 2009*; *Kidane et al., 2013*). In recent years, the number of experimentally characterized LC8 partners has risen to more than 40, and prediction methods indicate that dozens more may specifically bind LC8 (*Rapali et al., 2011b*). Gaining insight into how LC8 interacts with partner proteins, and how LC8 levels in the cell are balanced, is therefore paramount to understanding the regulation of many cellular processes.

A distinctive feature of ASCIZ is the high number of LC8 recognition motifs within its C-terminal domain. Although some LC8 partners have multiple recognition motifs (*Dunsch et al., 2012*; *Fejtova et al., 2009*; *Gupta et al., 2012*; *Stelter et al., 2007*) (Nucleoporin Nup159 has 5, Chica and Bassoon each has 3), human ASCIZ contains 11 functional LC8 binding sites (*Rapali et al., 2011a*), the most by far of any partner protein identified to date. This enrichment in LC8-binding sites is conserved throughout the animal kingdom, underscoring the importance of multiple motifs in ASCIZ function (*Figure 1d*). Cell culture transcription assays demonstrate that ASCIZ regulates LC8 transcription via a system of negative autoregulation, for which the mechanism is not well understood. Disruption of the ASCIZ/LC8 interaction via mutation of the TQT sites results in an increased level of LC8 transcription, while overexpression of LC8 decreases transcription (*Jurado et al., 2012a*). This observation led to the hypothesis that ASCIZ acts as a sensor for cellular LC8 and regulates LC8 transcription levels according to cellular needs (*Jurado et al., 2012a*). As LC8 expression levels vary among tissue types (*Chintapalli et al., 2007*) and LC8 overexpression enhances the survival and proliferation of breast cancer cells in culture (*Vadlamudi et al., 2004*), regulation of LC8 levels is critical for cellular health and homeostasis. However, while high levels of LC8 inhibit ASCIZ transcriptional activity (*Jurado et al., 2012a*), it is not known how this activity is controlled at the molecular level nor the requirement for multiple binding sites.

In this work, we use a combination of biophysical, structural, and molecular biology tools to explore the relationship between ASCIZ multivalency and LC8 transcription. We show that human and *Drosophila* ASCIZ bind to multiple LC8 dimers simultaneously in both a positively and negatively cooperative fashion, enabling the formation of a dynamic equilibrium of complexes, of which low occupancy intermediates are highly populated. We propose that this dynamic ensemble of complexes is important for transcriptional regulation and validate the main aspects of our hypothesis via transcriptional assays with human ASCIZ. These observations support a novel model of autoregulation, whereby ASCIZ engages in a dynamic equilibrium of multivalent interactions that tune the level of ASCIZ transcriptional activity.

## Results

### Unbound ASCIZ is a primarily disordered monomer

The 45 kDa *Drosophila* ASCIZ protein, dASCIZ, is predicted to contain four Zn-finger motifs at the N-terminus followed by a 243-amino acid long region of intrinsic disorder. The disordered region has six predicted LC8 recognition sites identified by a canonical TQT motif (*Figure 2a*, dark blue bars): QT1 (residues 251–262), QT2 (274–285), QT4 (323–334), QT5 (340–351), QT6 (354–365), and QT7 (374–385). QT3 (285–296) lacks the TQT residues but is identified experimentally as an LC8 recognition site in this work (below). Purification of full-length dASCIZ is impeded by poor expression levels and insolubility, and therefore we designed and produced constructs corresponding to the

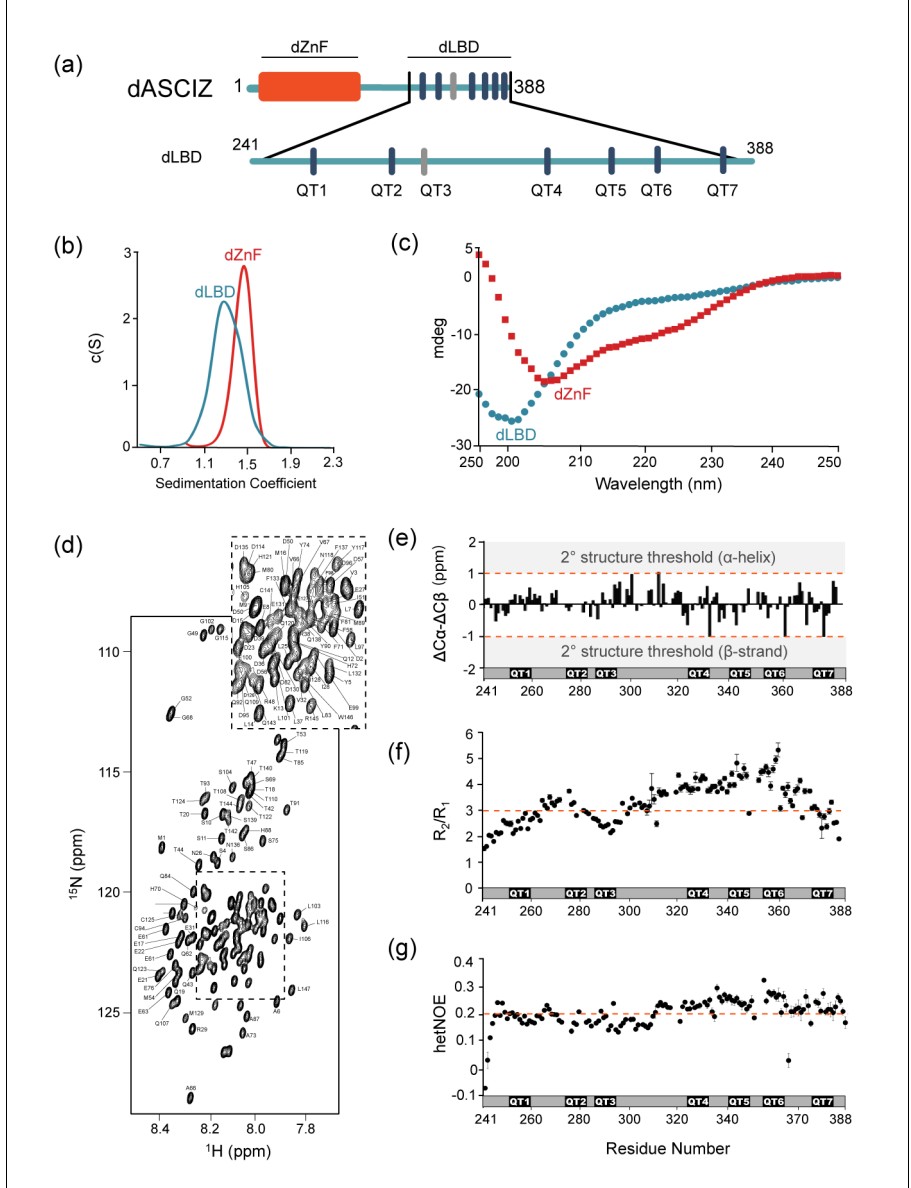

**Figure 2.** Domain structures of dASCIZ. (a) Domain structure of dASCIZ, showing the dZnF domain (red) and 7 LC8 binding motifs in its C-terminal domain (blue). Dark blue bars indicate predicted TQT motifs and gray bars indicate the TMT motif (QT3) identified in this study. (b) Sedimentation velocity analysis of the dZnF domain (red), collected at 10℃, and the dLBD (blue), collected at 25℃. (c) Far UV CD spectrum of the ZnF domain (red squares) and the dLBD (blue circles), both collected at 10℃. (d) [$^{15}$N−$^{1}$H]-BEST-TROSY spectrum at 850 MHz showing backbone assignments for 133 non-proline residues. Unassigned peaks correspond to the two additional N-terminal residues from the expression vector. The spectrum was recorded at 10℃. (e) A plot of secondary chemical shift differences versus residue number. ΔCα and ΔCβ values were calculated by subtracting the random coil chemical shifts (*Tamiola et al., 2010*) from the experimentally determined. ΔCα − ΔCβ values < ±1.0 ppm are considered not significant. (f) Plots of $R_2/R_1$ and (g) heteronuclear NOE values measured at 10℃ indicate high level of disorder. A dotted line is placed at the average value to aid in visualization. Segments corresponding to LC8 recognition motifs, QT1, QT2, QT3, QT4, QT5, QT6, and QT7 are shown. Additional structural characterization of the dLBD by circular dichroism and gel filtration chromatography is shown in *Figure 2—figure supplement 1*.

DOI: https://doi.org/10.7554/eLife.36258.005

The following source data and figure supplement are available for figure 2:

**Source data 1.** Tables containing the NMR data that were used to generate the graphs in *Figure 2e–g*.

*Figure 2 continued on next page*

*Figure 2 continued*
DOI: https://doi.org/10.7554/eLife.36258.007
**Figure supplement 1.** dLBD C-terminus is transiently compact.
DOI: https://doi.org/10.7554/eLife.36258.006

zinc finger domain, dZnF (residues 1–156, red bar *Figure 2a*), and the LC8-binding domain, dLBD (residues 241–388, *Figure 2a*).

Sedimentation velocity analysis of the dZnF and dLBD (*Figure 2b*) indicates that each is a monomer in solution with molecular weights of 17.4 kDa and 18 kDa, respectively (theoretical MW 17.6 kDa and 17 kDa). The CD spectrum of dZnF shows a large negative ellipticity at 208 nm and a small negative ellipticity at 222 nm (*Figure 2c*, red) indicative of a mix of alpha helices and loops, similar to CD spectra of other ZnF proteins (*Ezomo et al., 2010*). The dLBD CD spectrum has a large negative ellipticity at 200 nm, indicating that it is primarily disordered (*Figure 2c*, blue). From 5D NMR experiments, backbone assignments for 90% of the 148 residues in dLBD were obtained (*Figure 2d*). A high level of disorder in dLBD is revealed by the limited amide proton chemical shift dispersion in $^{15}$N HSQC spectra (*Figure 2d*), and a lack of secondary structure preference is further supported by small $\Delta C\alpha$-$\Delta C\beta$ chemical shift differences from random coil values (*Figure 2e*). Together these data demonstrate that dASCIZ contains an N-terminal structured domain as well as a long intrinsically disordered domain, and constructs of each domain are monomeric in solution.

Local dynamics of dLBD were assessed by NMR measurement of the $^{15}$N longitudinal ($R_1$), transverse ($R_2$) relaxation, and $^1$H-$^{15}$N heteronuclear NOEs. $R_2/R_1$ values range from 1.5 to 5.3 with an average of 3.3 (*Figure 2f*). Relatively higher $R_2/R_1$ values for residues 321–363 suggest motional restriction in this region. Heteronuclear NOE values measured at 10°C are very low overall, with values ranging from −0.1 to 0.3, but are also slightly higher for residues 321–363 (*Figure 2g*). Together, the $R_2/R_1$ and heteronuclear NOE values imply that dLBD is highly flexible with slight motional restriction in its C-terminal half.

The difference in flexibility between the N-terminus and C-terminus was validated by generating shorter constructs of the dLBD that include the first three (QT1-3), two internal sets of three (QT2-4 and QT4-6), and last four binding sites (QT4-7) (*Figure 3a*). All constructs are of a similar size, varying from 68 to 84 residues in length. Circular dichroism demonstrates that the C-terminal constructs, QT4-6 and QT4-7, are slightly more ordered than N-terminal constructs, QT1-3 and QT2-4 (*Figure 2—figure supplement 1a*). Size-exclusion chromatography supports this result, as the QT4-6 and QT4-7 constructs elute later, indicating that they are more compact than the N-terminal constructs (*Figure 2—figure supplement 1b*).

## Identification and binding affinities of LC8 recognition motifs in ASCIZ

Isothermal titration calorimetry (ITC) experiments on dASCIZ constructs were applied to identify the number of recognition motifs and provide estimates of their overall binding affinity for LC8. A range of constructs containing three to seven recognition motifs were tested for their binding to LC8 and all display a single binding step (*Figure 3*). Thus the measured $K_d$ and stoichiometry are 'effective' values that present an overall simplified picture of a much more complicated complex assembly process.

The full length dLBD binds LC8 with a dLBD:LC8 stoichiometry of 1:7 (two chains of dLBD for 7 LC8 dimers) and an overall $K_d$ of 1.4 µM (*Figure 3a*, *Table 1*), suggesting that an additional non-TQT ASCIZ site binds LC8. A plausible candidate is a TMT motif corresponding to residues 285–296 (designated QT3 in *Figure 3a*). To confirm the functionality of this motif, ITC binding of LC8 was measured for constructs QT1-3, QT2-4, QT4-6, and QT4-7 (*Figure 3b*). QT1-3 and QT2-4 contain the TMT binding motif, and both bind LC8 with a stoichiometry of 3, demonstrating that this TMT motif is the seventh LC8 recognition site (*Figure 3b*, *Table 1*). Each of the other two constructs bind LC8 with the stoichiometry expected from the number of TQT binding motifs. Interestingly, construct QT4-6, containing recognition sites 4–6, binds LC8 with a $K_d$ of 1.0 µM, a slightly higher overall affinity than the full dLBD construct. QT4-7, on the other hand, binds LC8 with a 1.6 µM affinity and is slightly more entropically disfavored than either QT4-6 or dLBD (*Table 1*). The overall $K_d$ values of QT1-3, QT2-4, and QT4-7, are 2.4 µM, 4.1 µM, and 1.6 µM, respectively.

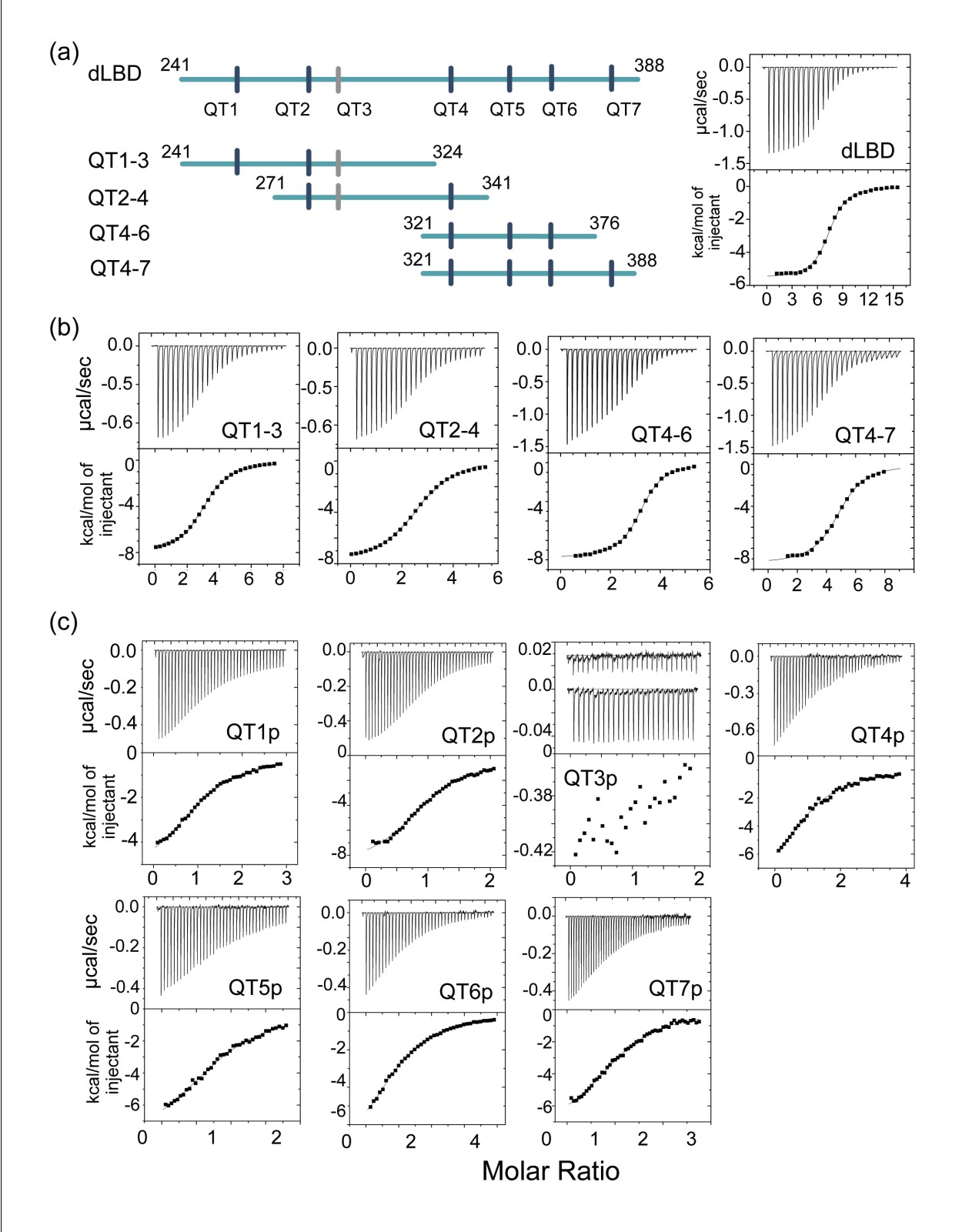

**Figure 3.** LC8-dLBD interactions monitored by ITC. (a) Construct schematics of the dLBD, QT1-3, QT2-4, QT4-6, and QT4-7 are shown, along with the locations of each TQT motif. A representative isothermal titration calorimetry thermogram of LC8 with the dLBD is shown on the right. (b) Representative thermograms of LC8 with constructs corresponding to QT1-3, QT2-4, QT4-6, and QT4-7. (c) Representative isothermal titration plots of LC8 with peptides corresponding to QT1p, QT2p, QT3p, QT4p, QT5p, QT6p, and QT7p. Since the interaction with QT3p is too weak, the heat

*Figure 3 continued on next page*

*Figure 3 continued*
exchange with buffer is plotted on top to show the difference between QT3p with LC8 and buffer. Higher concentrations were not possible due to poor solubility of the QT3p. Data were fit to a single site binding model using Origin software.
DOI: https://doi.org/10.7554/eLife.36258.008

Given the small differences in binding affinity among the shorter dLBD constructs, we asked if any individual site binds LC8 with higher affinity than all others. A series of 14–15 amino acid peptides were synthesized, each corresponding to one of the seven recognition motifs, and their LC8 affinity was measured (*Figure 3c*, *Table 2*). The $K_d$ values for all peptides indicate surprisingly weak affinity. QT3p interaction with LC8 is not even detected at 30 µM LC8. Intriguingly, the QT4-6 construct has a slightly higher binding affinity than the other constructs, while the QT4, QT5, and QT6 peptides display among the lowest affinity as individual peptides. These results indicate that positive cooperativity enhances LC8 binding to neighboring recognition motifs.

## dLBD and LC8 Form Dynamic, Low-Occupancy Complexes

Since ITC experiments on the full length dLBD and smaller constructs identify 7 LC8 recognition sites, we sought to establish the size of the dLBD:LC8 complex at varying LC8 concentrations by analytical ultracentrifugation. The dLBD was titrated with increasing concentrations of LC8 and complex formation was assessed at dLBD:LC8 molar ratios 1:1, 1:3, 1:6, and 1:10. Plots of the continuous size distribution, c(S), vs. sedimentation coefficient (*Figure 4a*) show that titration at sub-saturating concentrations of LC8 results in a broad peak that is likely an equilibrium mixture of complexes with varying LC8 occupancy in exchange with each other and with free dLBD. At a saturating concentration of LC8 (1:10 ratio), a high occupancy complex whose size (7.5 s) approximately corresponds to a fully bound complex (197 kDa) is clearly evident. Contrary to expectations, a low stoichiometry complex (5 s, approx. 114 kDa) is even more highly populated. A high frictional ratio ($f/f_0 \sim 1.6$) indicates an elongated complex, and the molecular weight of this low occupancy complex corresponds roughly to a 1:3 complex, although its broadness, and the approximate nature of the molecular weight determination, indicate that multiple species are present. It is likely that the low occupancy peak, which is roughly twice the intensity of the high occupancy peak, is a heterogeneous mixture of 1:2, 1:3 and 1:4 complexes.

In summary, while high occupancy complexes are evident in AUC profiles, consistent with the ITC results, these complexes are in equilibrium with many smaller sub-saturated species, the most populated of which is a mixture of 1:2-1:4 complexes of dLBD:LC8. The low occupancy complexes are favored, relative to higher occupancy complexes, even in samples having a large excess of LC8.

The presence of stable, low occupancy complexes is supported by small angle X-ray scattering (SAXS) data. A sample composed of dLBD and a large excess of LC8 was injected into an in-line size-exclusion chromatography system and X-ray scattering data were collected for the largest peak. Guinier analysis of the data indicates a monodisperse sample suitable for further analysis (*Figure 4— figure supplement 1a*). The distance distribution function suggests a moderately compact structure for dLBD:LC8 complexes, with $D_{max} = 240$ Å, and a molecular weight of roughly 110 kDa, confirming the presence of an unresolved mixture of dLBD:LC8 complexes with stoichiometries ranging from 1:2-1:4 (*Figure 4—figure supplement 1b*). Additionally, a Kratky plot of the scattering data

**Table 1.** Thermodynamic parameters of dASCIZ-LC8 interactions.

| Construct | N | Overall $K_d$ (µM) | Overall ΔH (kcal/mol) | Overall TΔS (kcal/mol) | Overall ΔG (kcal/mol) |
|---|---|---|---|---|---|
| dLBD | 7.3 | 1.4 ± 0.1 | −5.3 ± 0.2 | 2.7 ± 0.4 | −8.0 ± 0.4 |
| QT1-3 (241–324) | 3.2 | 2.4 ± 0.1 | −8.1 ± 0.4 | −0.4 ± 0.6 | −7.7 ± 0.4 |
| QT2-4 (271–341) | 2.7 | 4.1 ± 0.2 | −7.9 ± 0.4 | −1.6 ± 0.5 | −6.3 ± 0.3 |
| QT4-6 (321–376) | 3.0 | 1.0 ± 0.1 | −10.0 ± 0.5 | −1.8 ± 0.6 | −8.2 ± 0.4 |
| QT4-7 (321–388) | 4.0 | 1.6 ± 0.4 | −10.0 ± 0.5 | −2.1 ± 0.6 | −7.9 ± 0.4 |

DOI: https://doi.org/10.7554/eLife.36258.009

**Table 2.** Thermodynamic parameters of peptide-LC8 interactions.

| Peptide | Peptide Sequence*,† | N | $K_d$ (µM) | ΔH (kcal/mol) | TΔS (kcal/mol) | ΔG(kcal/mol) |
|---|---|---|---|---|---|---|
| QT1p | ymssQKLDMETQTEe | 1.1 | 14 ± 3.5 | −5.9 ± 0.3 | 0.7 ± 0.4 | −6.6 ± 0.3 |
| QT2p | ylapLLRDIETQTPd | 1.0 | 7 ± 0.4 | −9.2 ± 0.5 | −2.2 ± 0.6 | −7.0 ± 0.4 |
| QT3p | ytpdTRGDIGTMTDd | — | weak | ——— | ——— | ——— |
| QT4p | dlqTSAHMYTQTCd | 1.1 | 15 ± 0.8 | −8.7 ± 0.4 | −2.1 ± 0.5 | −6.6 ± 0.3 |
| QT5p | eelGLSHIQTQTHw | 0.9 | 11 ± 0.6 | −8.8 ± 0.4 | −2.0 ± 0.5 | −6.8 ± 0.3 |
| QT6p | wpdgLYNTQHTQTCd | 1.1 | 20 ± 1.0 | −8.6 ± 0.4 | −2.2 ± 0.5 | −6.4 ± 0.3 |
| QT7p | epdNFQSTCTQTRw | 1.1 | 10 ± 0.5 | −7.8 ± 0.4 | −0.9 ± 0.5 | −6.9 ± 0.3 |

*the 10-amino acid LC8 binding motif is capitalized

†non-native residues added to the N-terminus of each peptide to increase solubility or improve concentration determination are underlined

DOI: https://doi.org/10.7554/eLife.36258.010

indicates that the dLBD:LC8 complex is a mix of globular domains and intrinsically disordered chains, consistent with low occupancy complex structures (*Figure 4—figure supplement 1c*).

Native gel electrophoresis titration of dLBD with LC8 corroborates the presence of a mixture of complexes (*Figure 4b*). When unbound, dLBD and LC8 each migrate as a single band. When LC8 is added to dLBD at a 1:1 molar ratio, the band for free LC8 disappears, and the complexes formed migrate above LC8, as a diffuse band likely corresponding to two dLBD chains bound to two or more LC8 dimers (since some free dLBD persists). As the molar ratio is increased, the diffuse upper band becomes a dark smear, free LC8 accumulates in a pronounced dark band, and the free dLBD band disappears. We think the most likely explanation is that the decreasing mobility of the upper edge of the smear indicates increasing sizes of the complexes formed. Unbound LC8 is clearly visible at ratios ≥ 1:4, indicating that a pool of free LC8 accumulates even at conditions well below LC8 saturation of dLBD.

We similarly assessed the gel mobility of complexes formed by LC8 and the LC8-binding domain of human ASCIZ (hLBD), which contains 11 TQT motifs (*Rapali et al., 2011a*) (*Figure 1d*). Very similar behavior is observed for hLBD, although the gel mobility of hLBD bands is much lower than the mobility of dLBD bands due to molecular sieving of the much larger hLBD (53 kDa) compared to dLBD (17 kDa). Molecular sieving is the dominate effect on hLBD mobility as both hLBD and dLBD are disordered and highly extended and have a pI of ~4. As LC8 is added to hLBD, the free hLBD band disappears and a lower mobility smear becomes increasingly evident. Decreasing mobility of the upper smear indicates increasing sizes of the complexes formed, while the appearance of a bands migrating the same as free LC8 at ratios ≥ 1:4 indicates the presence of a pool of free LC8 well below LC8 saturation of hLBD. Together the gel titration data for dLBD and hLBD suggest that hLBD:LC8 complexes form a dynamic ensemble with varying levels of LC8 occupancy in which lower occupancy forms are favored.

Titration of the hLBD with LC8 by size-exclusion chromatography (*Figure 4d*) supports our interpretation of the native gel experiments in *Figure 4c*. Here, the amount of hLBD is held constant, and increases in peak intensity are due to effects on hLBD complexes from increasing LC8. Both LMW and HMW complexes form at the first titration point and increase in population as more LC8 is added. A low occupancy intermediate, labeled LMW, is clearly evident at the lowest molar ratio of 1:3, and persists at the same elution volume even at the highest molar ratios. Higher occupancy species, HMW, are also apparent at a ratio of 1:3 and become increasingly distinct as LC8 is increased. The peak corresponding to excess LC8 is discernable at a ratio of 1:7, and steadily increases as LC8 is added. Taken together, the data in *Figure 4* are consistent with the explanation that a low occupancy form of the hLBD:LC8 complex is favored even with a large excess of LC8. A minor population of high occupancy complexes, present even at the lowest molar ratio, increases with increasing LC8 and is in equilibrium with the LMW species and with free LC8, and therefore with each other. Both *Drosophila* and human LBD exhibit this dynamic behavior, suggesting it is a conserved feature of the ASCIZ:LC8 interaction.

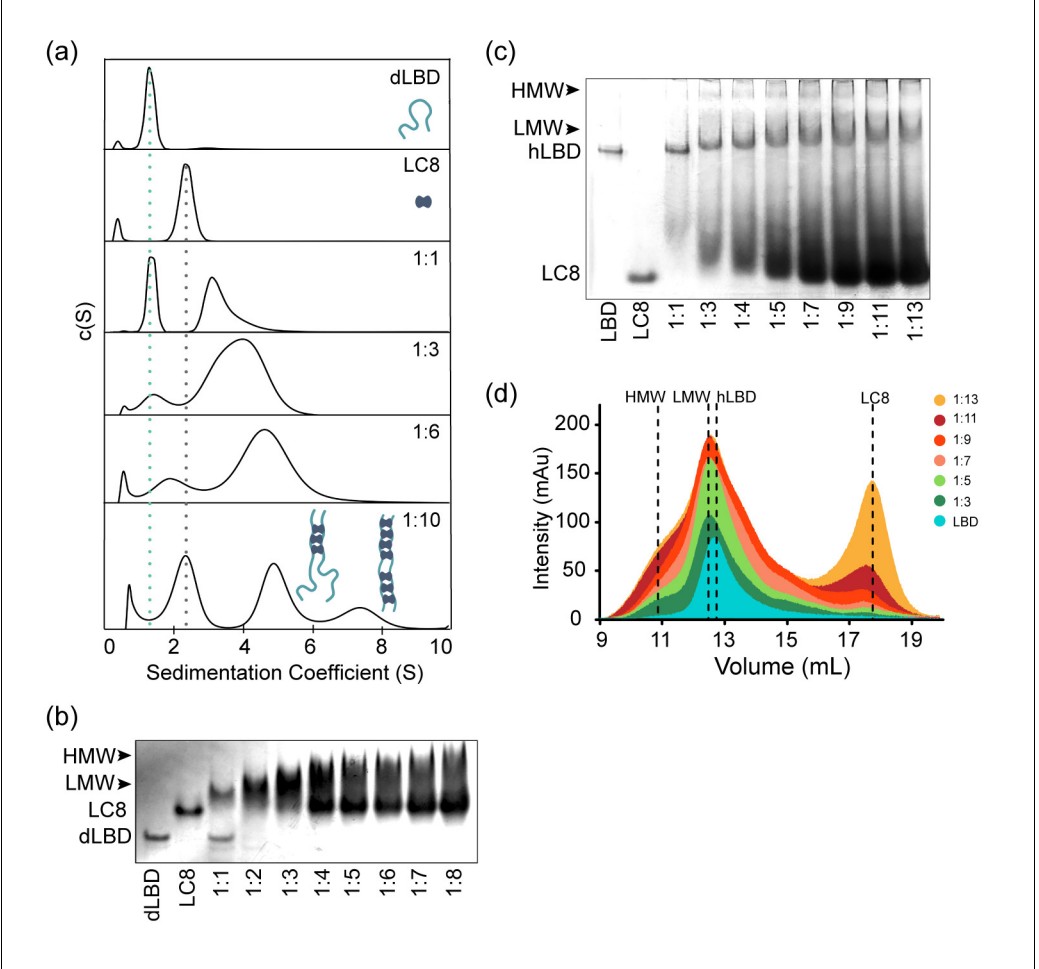

**Figure 4.** ASCIZ and LC8 form a dynamic complex with low occupancy intermediates. (**a**) Representative *c(S)* distributions obtained by sedimentation velocity are shown for the dLBD, LC8, and increasing molar ratios of the dLBD: LC8; 1:1, 1:3, 1:6, and 1:10. The standardized sedimentation coefficients ($s_{20,w}$) of unbound LC8 and dLBD determined from sedimentation velocity are 2.42 s (frictional ratio, $f/f_0$ = 1.21, consistent with a globular protein) and 1.34 s ($f/f_0$ = 1.84, consistent with an asymmetric or unfolded protein), respectively. Calculation of the molecular weight from the sedimentation coefficient and frictional ratio gave masses of 23 kDa and 18 kDa respectively, closely matching their theoretical masses of 24 kDa for LC8 dimer and 17 kDa for dLBD monomer. Cartoon depictions of the ASCIZ dLBD (light blue) and LC8 (dark blue) are shown to aid in visualization of the complexes formed at a 1:10 ratio. (**b**) Native gel titration of dLBD with LC8. An increasing concentration of LC8 was added to a constant amount of dLBD, from a molar ratio of (dLBD:LC8) 1:1 to 1:8. As more LC8 is added, the complex migrates more slowly and excess LC8 appears at 1:4. (**c**) Native gel titration of hLBD with LC8. As an increasing concentration of LC8 is added to a constant amount of hLBD. Arrows indicate the locations of the low molecular weight (LMW) and high molecular weight (HMW) complex. (**d**) Titration of the hLBD with LC8, monitored by size exclusion chromatography on a Superdex 200 gel filtration column. The concentration of hLBD is held constant and an increasing amount of LC8 is added, from a molar ratio of 1:3 to 1:13. Peaks corresponding to free hLBD, free LC8, low molecular weight complex (LMW) and high molecular weight (HMW) complex are labeled. Additionally, SAXS data for the LMW dLBD:LC8 complex is shown in *Figure 4—figure supplement 1*.

DOI: https://doi.org/10.7554/eLife.36258.011

The following figure supplement is available for figure 4:

**Figure supplement 1.** SAXS data of dLBD:LC8 complex.

DOI: https://doi.org/10.7554/eLife.36258.012

## Structure and distribution of LBD:LC8 complexes visualized by single particle EM

In order to visualize the various oligomeric states of ASCIZ-LC8, we analyzed electron microscopy data of dLBD and hLBD under saturating concentrations of LC8. As a positive control, and for validation of EM conditions, similar experiments were carried out with complexes of Nucleoporin159

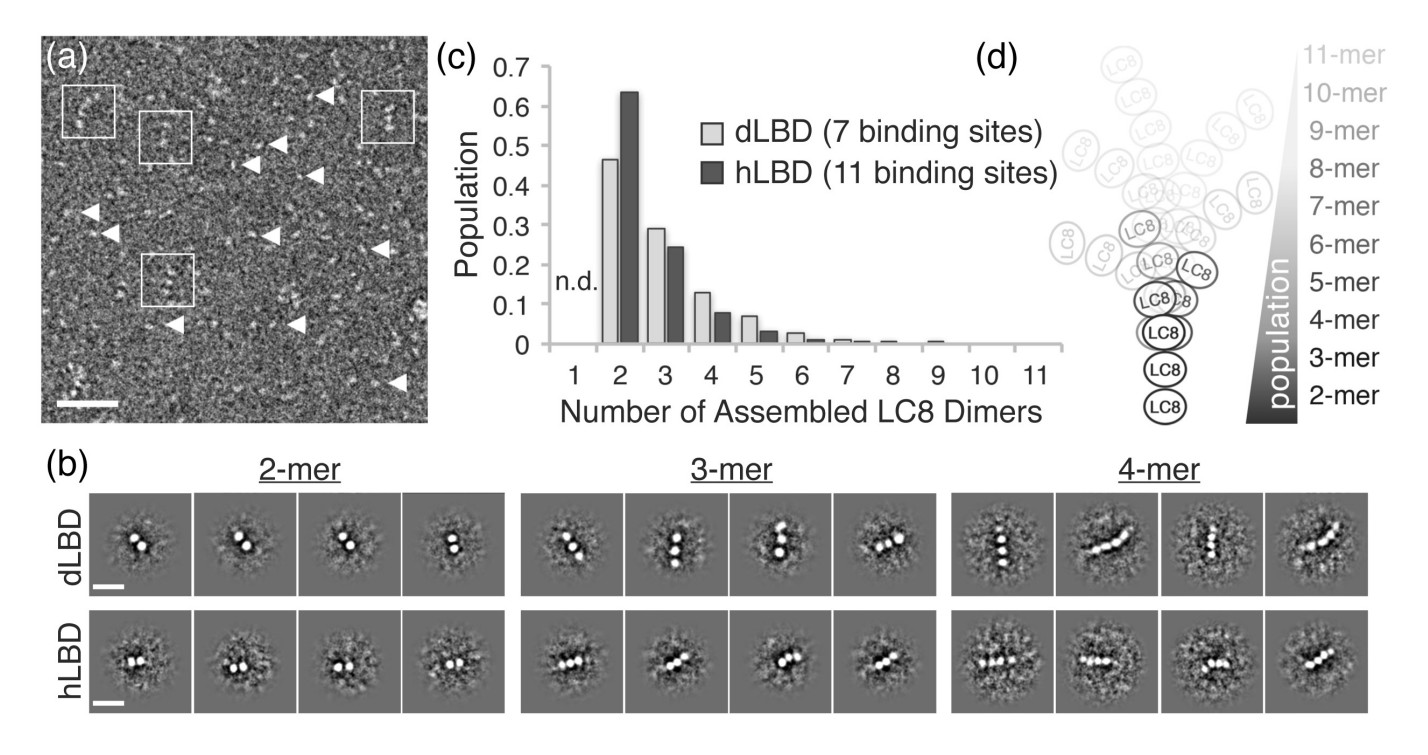

**Figure 5.** dLBD:LC8 and hLBD:LC8 complexes visualized by negative stain electron microscopy. (a) Representative micrograph of negatively stained hLBD:LC8 complexes. Identified oligomeric complexes are boxed. Non-oligomeric LC8 dimers are indicated by arrowheads. Scale bar = 100 nm. dLBD:LC8 micrographs had the same appearance (*not shown*). (b) Representative 2D projection averages of (*top*) dLBD:LC8 oligomers and (*bottom*) hLBD:LC8 oligomers. Only low-occupancy oligomers with 2–4 LC8 dimers (2mer – 4mers) were successfully averaged. Higher-occupancy oligomers were identified in raw micrographs (*Figure 5—figure supplement 1*), but were not averaged due to low population and/or high degree of conformational flexibility. Scale bar = 20 nm. (c) Histogram showing the normalized population distribution of LC8 occupancy in complexes formed with dLBD (*grey*) and hLBD (*black*), identified from raw micrographs. The population of complexes formed with a single LC8 dimer were not determined (*n.d.*) (d) Illustration representing the distribution of LC8 occupancy and conformational flexibility observed in hLBD:LC8 complexes.
DOI: https://doi.org/10.7554/eLife.36258.013

The following figure supplement is available for figure 5:

**Figure supplement 1.** Single particle images of dLBD:LC8 and hLBD:LC8 complexes.
DOI: https://doi.org/10.7554/eLife.36258.014

(Nup159), another intrinsically disordered protein with multiple LC8 binding sites (*Stelter et al., 2007*). Nup159:LC8 complexes were clearly visualized as a linear array of 5 stacked densities of LC8, as previously reported (*Stelter et al., 2007*) (data not shown), and consistent with the conclusions of Nup159:LC8 biophysical solution experiments (*Nyarko et al., 2013*). In contrast, despite the similar overall affinity (Nup159-LC8 $K_d$ = 2.9 μM) (*Nyarko et al., 2013*), in negative stain images of dLBD:LC8 and hLBD:LC8 complexes, the vast majority of complex species appear dissociated on the grid (*Figure 5a, arrow heads*). However, a few observable complexes could be clearly resolved from raw micrographs, identified as linear stacks of punctate densities, akin of beads on a string (*Figure 5a, squares*), similar to images of Nup159 bound to LC8 (*Stelter et al., 2007*). Furthermore, although dLBD contains seven and hLBD 11 LC8 binding sites, the vast majority of complexes observed by EM appeared to be of low LC8 occupancy.

Reference-free two-dimensional (2D) classification routines were carried out on datasets of ~2000 single particle images of dLBD:LC8 complexes and ~1000 particles of hLBD:LC8 complexes extracted from ~300 and 200 micrographs, respectively. These produced 2D projection averages for dLBD:LC8 and hLBD:LC8 oligomers displaying complexes formed with 2–4 stacked densities, corresponding to LC8 dimers, deduced from the dimensions of the averaged bead-like densities (~4 nm diameter) (*Benison et al., 2007*; *Stelter et al., 2007*). Complexes with three or more LC8 dimers displayed significant conformational flexibility in 2D class averages (*Figure 5b*) and in the single-

particle images (*Figure 5a* and *Figure 5—figure supplement 1*). The extent of conformational variability is consistent with ~10–20 Å spacing measured between LC8 densities, and the intrinsic flexibility of the IDP duplex chain separating the neighboring LC8 TQT recognition motifs.

The formation of higher-order oligomers appeared relatively rare in comparison to the low occupancy complexes. The scarcity of higher order complexes, coupled with the intrinsic conformational heterogeneity, precluded our ability to obtain 2D class averages of the high-occupancy complexes. To overcome this limitation, statistical analysis describing the distribution of oligomeric states was obtained by hand-selection and classification from single-molecule images (*Figure 5—figure supplement 1*). Both dLBD:LC8 and hLBD:LC8 complexes form an ensemble of structures, displaying an exponential distribution with low-occupancy states (*i.e.* 2–4 stacked LC8 dimers) being most abundant (*Figure 5c,d*). Density corresponding to the IDP duplex chain cannot be resolved by negative stain EM, therefore complexes formed with a single LC8 dimer were not included in this analysis, as they could not be distinguished from unbound LC8 dimers.

Together, this analysis shows dLBD and hLBD form dynamic assemblies with LC8 that favor low occupancy states (*Figure 5d*). Although uncommon, high-occupancy and fully-formed complexes of dLBD:LC8 (1:7 ratio) could be identified from the raw single particle images (*Figure 5—figure supplement 1*), further confirming the stoichiometry obtained by our ITC studies. For the hLBD:LC8 dataset, complexes containing as many as 7–9 LC8 dimers could be distinguished from the single particle image data, while higher-order complexes containing 10–11 LC8 dimers were either not distinguishable or were simply absent under the limiting concentrations required for negative stain EM specimen preparation. Nevertheless, the remarkable similarity in distribution of oligomeric species formed by dLBD and hLBD, obtained under similar binding conditions, is consistent with the nearly equivalent overall LC8 affinity determined by ITC, and suggests that a conserved mechanism of negative cooperativity is used by ASCIZ to regulate the formation and distribution of higher-order LC8 assemblies.

## NMR titration of dLBD with LC8 identifies sites with modest preferential binding

If dLBD and LC8 form stable intermediate complexes with excess LC8, which of the seven recognition sites in *Drosophila* ASCIZ are preferentially bound? Notably, the linear assembly pattern of LC8 dimers observed by negative stain EM suggests an ordered (or quasi-ordered) sequence of assembly, apparently favoring neighboring TQT sites. However, the location of LC8 binding sites could not be resolved in these experiments. Therefore, to examine interactions between individual motifs and LC8 in the context of the full dLBD, we turned to NMR. As unlabeled LC8 is titrated into solutions of $^{15}$N-$^{13}$C- labeled dLBD, changes in NH peak intensities can be measured in 3D HNCO spectra recorded for dLBD:LC8 molar ratios of 1:0.25, 1:1, 1:2, 1:5, 1:8. As LC8 concentration increases, a corresponding decrease in dLBD peak intensity is observed (*Figure 6a*). At a molar ratio of 1:5, less than 10% of the original peak intensity remains at all seven LC8 binding sites. At a ratio of 1:8, all dLBD peaks completely disappear except for peaks corresponding to eight N-terminal residues (241-248), indicating that all TQT sites have been occupied to some degree. The absence of peaks for bound dLBD is attributed to line broadening associated with intermediate exchange processes and/or faster transverse relaxation as a result of increased complex size. Therefore, we consider a decrease in peak intensity as a measure of increased complex formation.

Notably, peaks at the C-terminal half of the protein, QT4-7, decrease more quickly than peaks at the N-terminal half, implying that LC8 preferentially occupies these motifs. To confirm this observation and to obtain titration information for the missing residues in this region, we performed similar experiments on the smaller QT2-4 and QT4-6 constructs. QT2-4 was chosen to further validate LC8 binding to the QT3 motif that as an individual peptide showed weak binding by ITC, and QT4-6 was chosen because it has the highest LC8 binding affinity (*Figure 3b*). Further, the two constructs share the QT4 motif, allowing us to assess its affinity in two sequence contexts. Due to the smaller number of peaks for shorter constructs, HSQC spectra have a sufficiently high resolution to render an HNCO-based titration unnecessary. Unlabeled LC8 was titrated into $^{15}$N-labeled QT2-4 or QT4-6 and changes in peak intensity were analyzed (*Figure 6b,c*). As with dLBD, there is a gradual decrease in peak intensity as more LC8 is added to QT2-4 or to QT4-6 at molar ratios: 1:0.25, 1:1, 1:2, 1:3, and 1:4. Significantly, peak intensities in QT4-6 decrease at lower LC8 ratio than in QT2-4,

confirming the trend we observe in full-length dLBD. In QT4-6, nearly all peaks in the motif region disappear at a ratio of 1:2, while ~30% of peak intensity remains in QT2-4.

Furthermore, NMR titration of QT2-4 with LC8 confirms that QT3 is an LC8 binding motif. The peaks corresponding to QT3 decrease in intensity at the same rate as peaks corresponding to the QT2 and QT4 motifs. The peaks in the linker region (residues 305–320) decrease more slowly,

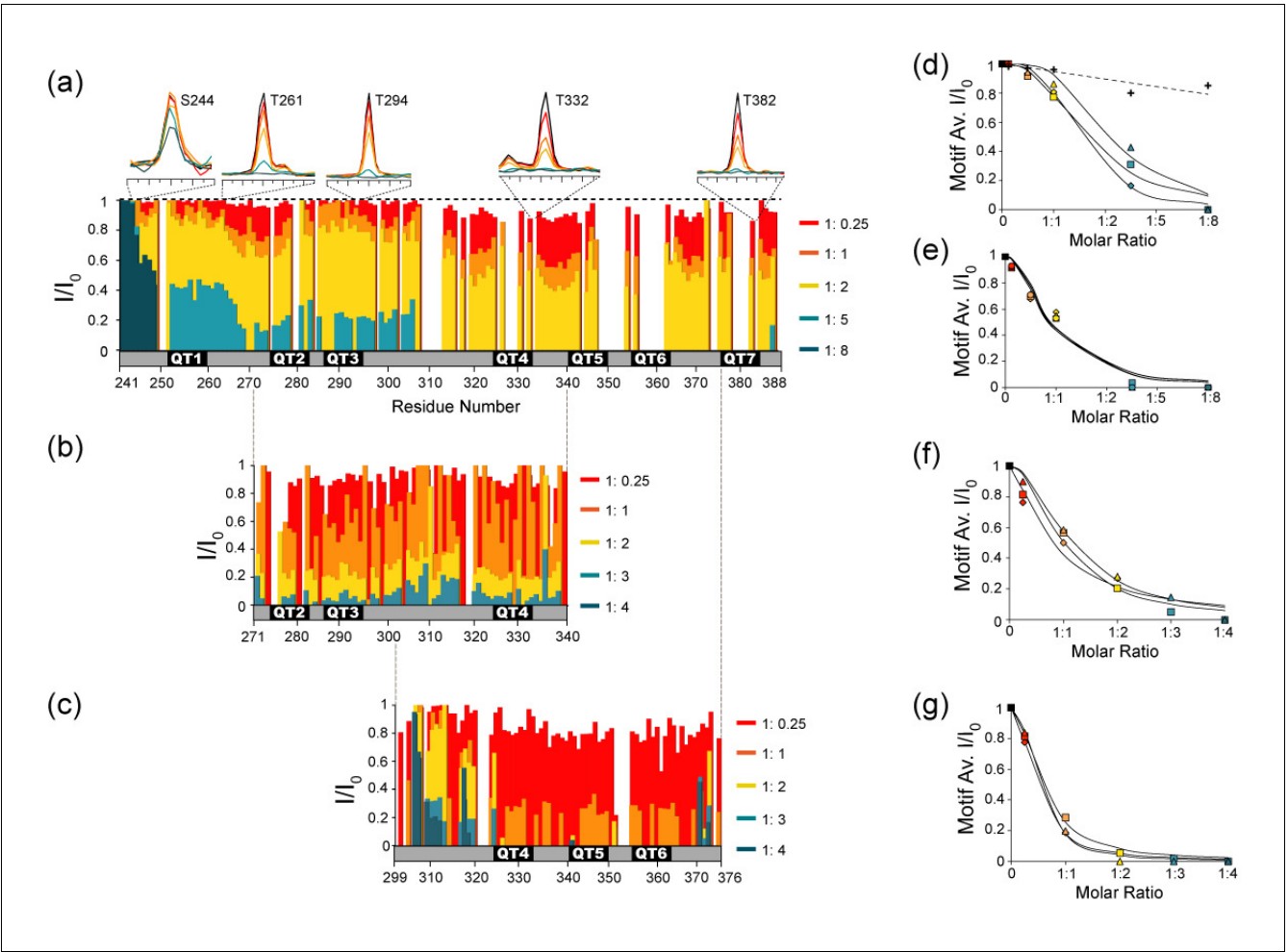

**Figure 6.** NMR titration of the dLBD with LC8. Relative intensities of non-proline NH peaks in $^{15}N$-$^{13}C$-HNCO spectra are shown for (a) dLBD, (b) QT2-4, and (c) QT4-6 titrated with LC8 at molar ratios of 1:0.25, 1:1, 1:2, 1:5, and 1:8. Peak intensities (I) are given relative to the intensity of the same peak in free peptide ($I_0$). Above (a) are 1D NMR slices of representative amino acids from TQT motifs QT1 (T261), QT3 (T294), QT4 (T332), QT7 (T382) and a control (S244). Representative slices of HNCO spectra from multiple titration points are shown in *Figure 6—figure supplement 1*. (d–g) The $I/I_0$ data are alternatively plotted as the average intensity ratio for each 10-amino acid motif versus the molar ratio of dLBD:LC8. For the dLBD complex, titration curves of individual motifs cluster in two groups having higher (d) or lower (e) average intensity ratios at the same dLBD:LC8 molar ratio. Similarly, for the two shorter constructs, plots of average $I/I_0$ for individual motifs cluster in two groups, motifs in QT2-4:LC8 having higher intensity ratios (f), and motifs in QT4-6:LC8 having lower intensity ratios (g). Motif designations in (d) and in (f) are: QT1 (triangle), QT2 (square), QT3 (diamond). Motif designations in (e) and (g) are: QT4 (triangles), QT5 (squares), QT6 (diamond), and QT7 (circle). Note that complete peak attenuation at any titration point is given a value of 0 (*Figure 6g*). In (d) values are shown for a negative control group (crosses, dotted line) comprised of those measurable at saturating LC8 concentration in the first eight amino acids, 241–249. In panels (d–g), solid curves are to guide the eye.
DOI: https://doi.org/10.7554/eLife.36258.015

The following source data and figure supplement are available for figure 6:

**Source data 1.** A table of the average $I/I_0$ values for each 10-amino acid motif in the dLBD construct, QT2-4 construct, and QT4-6 construct.
DOI: https://doi.org/10.7554/eLife.36258.017

**Figure supplement 1.** Representative HNCO slices of dLBD titration with LC8.
DOI: https://doi.org/10.7554/eLife.36258.016

indicating that they are not interacting with LC8, but merely experiencing the effects of a larger correlation time.

Plots of the average peak intensity ($I/I_0$) for each 10-amino acid motif in dLBD, and in each of QT2-4 and QT4-6 constructs clearly show the dichotomy in the pattern of peak attenuation (*Figure 6d–g*). In dLBD, the first three (*Figure 6d*) show a different titration pattern and weaker binding than the last four motifs (*Figure 6e*). This dichotomy is replicated in separate plots of the 3 QT motifs in each of the constructs QT2-4 and QT4-6; the average $I/I_0$ of motifs in QT2-4 (*Figure 6f*) drops to 0.2 at LC8 molar ratio 1:2, twice that observed for motifs in QT4-6 (*Figure 6g*) which reach the same $I/I_0$ at molar ratio 1:1. The apparently higher LC8 affinity of motifs 4–6 (*Figure 6e,g*), relative to motifs 2–4 (*Figure 6d,f*) is consistent with our ITC experiments (*Figure 3a*). The QT4 motif, common to both constructs, has a different rate of peak disappearance in each construct, suggesting that motif environment, not local sequence, determines its affinity. We conclude that the recognition motifs QT4-QT7 are the sites favored in stable low occupancy complexes.

In summary, the data in *Figure 6* showing peak attenuation across the whole sequence, even at low LC8 ratios, suggest population of an ensemble with all LC8 sites occupied to varying degree, but with clear preference for the C-terminal motifs. Preferential binding of the C-terminal motifs is additionally supported by ITC results, which show that LC8 binds to the QT4-6 construct with slightly higher affinity than the full-length dLBD (*Figure 3a,b*). NMR dynamics experiments also demonstrate that residues in the C-terminal motifs are slightly more ordered in comparison to the N-terminal motifs (*Figure 2f–g*, *Figure 2—figure supplement 1a*), which may explain the tighter LC8 binding to this region.

## Cell-Based assays show a gradient of transcriptional activity modulated by LC8 binding

To investigate how the number of bound LC8 molecules affects the transcriptional activity of ASCIZ, we turned our attention to the human protein whose transcriptional activity can be assayed in cell culture using an ASCIZ knockout mouse embryonic fibroblast cell line (*Jurado et al., 2012a*). Human ASCIZ has eleven LC8 recognition motifs that we have numbered 1 to 11 (*Figure 7a*). To prevent LC8 binding to specific ASCIZ motifs, TQT recognition motifs were mutated to AAA. Five human ASCIZ mutant constructs were generated: AAA1-4 with an AAA replacement at each TQT motifs 1–4, and similarly named mutant constructs of AAA8-11; AAA5-11; AAA1-4, 8–11; and AAA-all.

To confirm loss of LC8 binding in vitro for each TQT when replaced with AAA, identical mutations were made in ASCIZ hLBD constructs, named in the same fashion. To completely eliminate LC8 binding in the AAA-all construct, it was necessary to also mutate three SQT/VQT motifs in addition to the 11 TQT motifs identified in our ITC experiments and in pepscan experiments (*Figure 7a*, gray bars) (*Rapali et al., 2011a*). This suggests that human ASCIZ may contain additional binding motifs, as seen with the TMT motif in *Drosophila* ASCIZ. The effective affinity and stoichiometry of binding of LC8 to WT ASCIZ hLBD and each mutant hLBD was determined (*Figure 7b*, and *Table 3*). WT hLBD binds LC8 with an hLBD:LC8 ratio of 1:11 (two chains of ASCIZ to 11 LC8 dimers), and an overall $K_d$ value of 0.9 μM (*Table 3*). The four hLBD mutants each binds LC8 with the expected stoichiometry for the number of intact LC8 recognition motifs and with overall affinities in the range of 0.7–4.4 μM.

To assess the impact of these mutations on ASCIZ transcriptional activity, luciferase reporter assays were carried out using immortalized ASCIZ knockout mouse embryonic fibroblasts transiently transfected with the full-length WT or a mutant ASCIZ gene and a plasmid containing the LC8 promoter. The measured luciferase activity was normalized against *Renilla* luciferase (*Figure 7c*). Empty vector and the ΔZnF construct showed limited transcriptional activity compared to ASCIZ constructs. Most significantly, transcriptional activity of the AAA mutants can be ranked to form an activity gradient (*Figure 7c*) notable for a clear inverse relationship between their transcriptional activity and their affinity for LC8. While the differences between each construct are very small, the overall trend supports the hypothesis that affinity and transcriptional activity are correlated. The construct with the highest affinity for LC8 by ITC, AAA8-11, exhibits equal or slightly lower transcriptional activity than WT ASCIZ, while the construct with the lowest affinity for LC8, AAA1-4, 8–11, has 2.5x the activity of WT ASCIZ. The correlation between transcriptional activity and affinity for LC8 is also shown in *Figure 7d*. As the number of available binding sites decreases and $K_d$ correspondingly increases, transcriptional activity also increases.

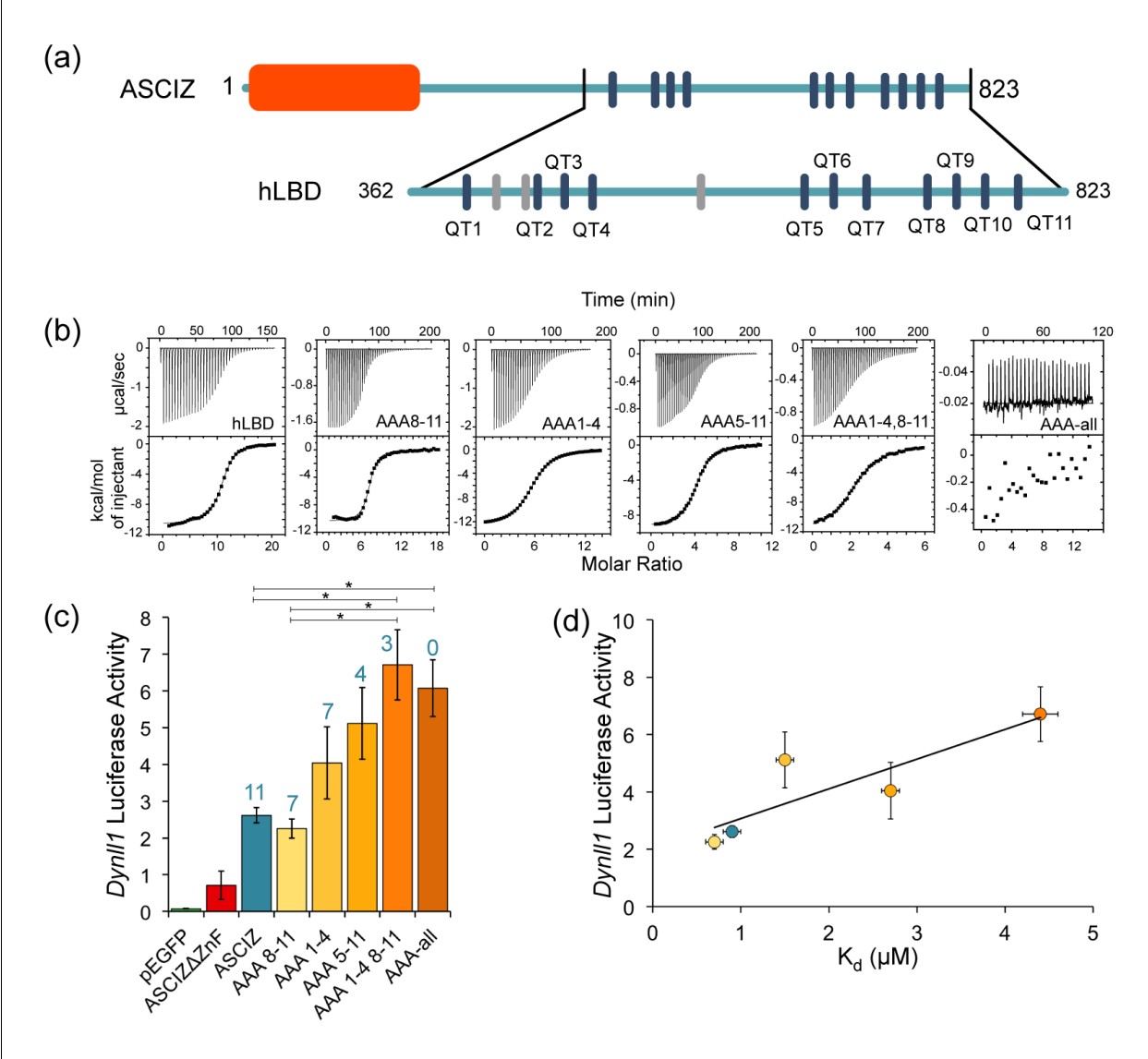

**Figure 7.** The number of LC8 recognition motifs tunes ASCIZ transcriptional activity. (a) Domain structure of human ASCIZ, showing 11 LC8 binding motifs as blue bars. Additional non-TQT motifs are shown as gray bars. (b) Shown are representative isothermal titration plots of LC8 with the hLBD and constructs with four (AAA1-4 and AAA8-11), seven (AAA5-11), eight (AAA1-4, 8–11), or fourteen (AAA-all) mutant LC8 binding sites. Data were fit to a single site binding model using Origin software. (c) Firefly luciferase reporter assays of ASCIZ knockout mouse embryo fibroblast cells transiently transfected with WT human ASCIZ (blue), a zinc finger deletion construct (red), and ASCIZ mutant constructs AAA1-4, AAA8-11, AAA5-11, AAA1-4, 8–11, or AAA-all (shades of orange), along with the *Dynll1* luciferase and *Renilla* luciferase vectors. The number of available motifs is indicated above each construct. Error bars are ±S.E. relative to *Renilla* luciferase as a control. Asterisks (*) indicate p-values less than 0.01. Data is the average of 2–4 independent experiments. Although the differences in transcriptional activity between each construct is small, the overall trend indicates an increase in transcriptional activity with a decrease in available binding motifs. A western blot depicting expression of the ASCIZ constructs is shown in *Figure 7— figure supplement 1*. (d) Binding affinity of each construct for LC8 is plotted against luciferase activity. Data points are colored according to (c). The AAA-all construct is excluded from the graph because it does not bind to LC8.

DOI: https://doi.org/10.7554/eLife.36258.018

The following source data and figure supplement are available for figure 7:

**Source data 1.** The raw data from the firefly luciferase reporter assays, shown for each ASCIZ construct.
DOI: https://doi.org/10.7554/eLife.36258.020

**Figure supplement 1.** Western blot of ASCIZ constructs
DOI: https://doi.org/10.7554/eLife.36258.019

**Table 3.** Thermodynamic parameters of ASCIZ-LC8 interactions.

| Construct | N | Overall $K_d$ (μM) | Overall $\Delta H$ (kcal/mol) | Overall $T\Delta S$ (kcal/mol) | Overall $\Delta G$ (kcal/mol) |
|---|---|---|---|---|---|
| hLBD | 11.2 | 0.9 ± 0.1 | −10.6 ± 0.5 | −2.4 ± 0.6 | −8.2 ± 0.4 |
| AAA8-11 | 6.7 | 0.7 ± 0.1 | −10.4 ± 0.5 | −2.0 ± 0.6 | −8.4 ± 0.4 |
| AAA1-4 | 6.6 | 2.7 ± 0.1 | −12.6 ± 0.6 | −5.0 ± 0.7 | −7.6 ± 0.4 |
| AAA5-11 | 4.2 | 1.5 ± 0.1 | −9.2 ± 0.5 | −1.3 ± 0.6 | −7.9 ± 0.4 |
| AAA1-4, 8–11 | 2.5 | 4.4 ± 0.2 | −12.2 ± 0.6 | −4.9 ± 0.7 | −7.3 ± 0.4 |

DOI: https://doi.org/10.7554/eLife.36258.021

One exception to this trend is the AAA-all construct with zero functional LC8 binding sites (AAA-all), which shows equal or somewhat lower activity than the construct with three intact sites (AAA1-4,8–11). As both the ZnF and LBDs of ASCIZ are monomeric in the absence of LC8, a plausible explanation for this effect could be that dimerization of ASCIZ by a minimal number of LC8 molecules is required for optimal binding to the *DYNLL1* gene promoter, or the binding of dimeric transcription co-activators to ASCIZ.

In summary, the data in *Figure 7* indicate that, in general, ASCIZ transcriptional activity appears to vary inversely with the number of LC8 recognition motifs and with binding affinity, and that fine tuning within this trend depends on which motifs are occupied and their specific dissociation constants.

## Discussion

A distinctive feature of ASCIZ is the high number of LC8 recognition motifs within its large, disordered C-terminal domain. Here we integrate multiple approaches to elucidate the structure, dynamics, thermodynamics, and hydrodynamics of the large disordered ASCIZ-LC8 complexes, that together reveal a new model by which ASCIZ can maintain stable pools of the hub protein LC8. We tested the main features of this model in cells using transcription activity assays which show a trend

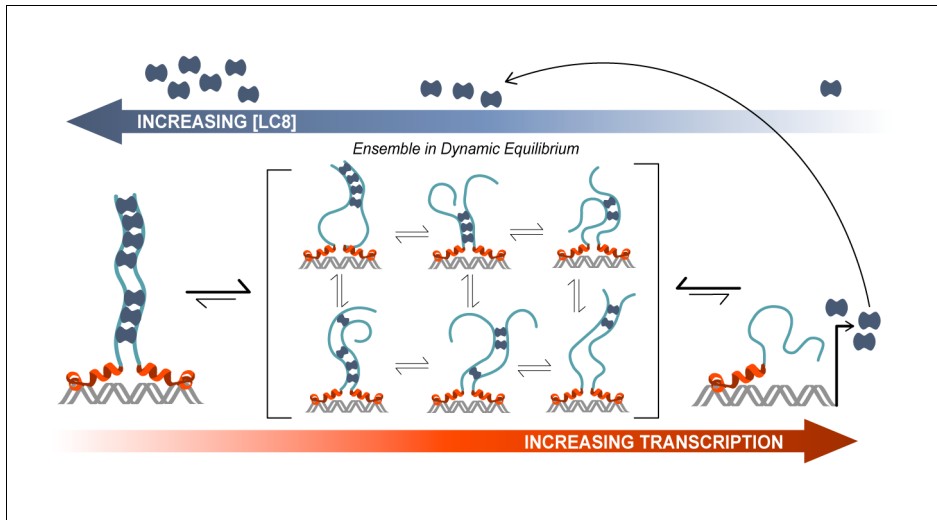

**Figure 8.** Model of ASCIZ regulation of LC8 transcription. A proposed model of LC8 transcriptional regulation is shown for dASCIZ, which also applies to the human protein. Free LC8 dimers (dark blue) bind to ASCIZ and modulate transcriptional activity. Transcriptional activity is lowest when ASCIZ recognition motifs are fully occupied by LC8 (left) and is highest when ASCIZ is bound to a small number of LC8 dimers or not bound to LC8 (right). A dynamic, low-occupancy complexes (center) are composed of ASCIZ bound to 2–4 LC8 dimers. LC8 that is produced upon ASCIZ transcription returns to the pool of free LC8.
DOI: https://doi.org/10.7554/eLife.36258.022

wherein mutant ASCIZ constructs with lower LC8 occupancy display higher transcriptional activity, while constructs with higher LC8 occupancy have lower activity. We propose that a dynamic ensemble of complexes is important for fine-tuning ASCIZ transcriptional activity, where stable, low occupancy complexes function to maintain a basal buffering transcription rate for LC8. A change in LC8 cellular concentration would shift this dynamic equilibrium to a higher or lower occupancy state without dramatically altering the level of transcription. As LC8 is an essential regulator of dozens of cellular processes, an ability to maintain an LC8 'buffer' is likely important for cellular homeostasis. Although many other transcription factors are regulated by multisite phosphorylation (*Holmberg et al., 2002*) or multiple binding events to different proteins (*Cantor and Orkin, 2002*; *Reményi et al., 2004*), we find no examples of activity tuned by multivalent binding to the gene product in a negative autoregulatory role, which underscores the novelty and potential impact of this study.

## ASCIZ is a new type of multivalent transcription factor

ASCIZ has three structural and functional features that together set it apart from other multivalent transcription factors.

(1) ASCIZ has an exceptionally long intrinsically disordered C-terminal domain compared to other intrinsically disordered transcription factors. Human ASCIZ contains an intrinsically disordered domain that is 600 amino acids long, or 73% of its overall sequence. The well-studied transcription factor p53, by contrast, is 40% disordered and the disordered regions are dispersed through three different regions of the protein rather than being concentrated on one terminus (*Laptenko et al., 2016*). Intrinsically disordered regions longer than 50 amino acids are considered to be of significant size for eukaryotic transcription factors (*Liu et al., 2006*). As intrinsic disorder is proposed to play an important role in regulating function (*Shammas, 2017*), it is possible that the length of the disordered domain enables a larger diversity of functions. Indeed, recent work has shown that the length of the intrinsically disordered domain can control transcriptional activity through 'energetic frustration', wherein opposing energetic couplings mediate the overall activity (*Li et al., 2017*).

(2) ASCIZ interactions with LC8 display *both* positive and negative cooperativity that together create a dynamic equilibrium of stable, low occupancy ASCIZ-LC8 complexes. ASCIZ binding to LC8 forms an IDP duplex scaffold (*Clark et al., 2015*) onto which other copies of LC8 or other dimeric partners can bind with higher affinity. The first two to four recognition motifs bind to LC8 with positive cooperativity, as evidenced by ITC experiments that show an enhancement in binding affinity from the presence of neighboring motifs (*Figure 3*, *Table 1*). Negative cooperativity regulates the formation and distribution of higher-order LC8 assemblies, as shown by the dominance of low occupancy complexes at saturating concentrations of LC8 (*Figures 4* and *5*). Negative cooperativity observed between low occupancy complexes and the fully occupied complex suggests that when the concentration of LC8 exceeds the buffering capacity of the low occupancy intermediates, the fully occupied complex is formed to switch off transcription.

A distribution of low occupancy dynamic complexes is a conserved feature of the ASCIZ:LC8 interaction. Evidence for a dynamic ASCIZ:LC8 ensemble comes from a combination of AUC, gel filtration, native gel electrophoresis, and negative stain electron microscopy data (*Figures 4–5*, *Figure 4—figure supplement 1*). For both *Drosophila* and human LBD constructs, addition of excess LC8 results in formation of stable low molecular weight (LMW) complexes and a minor population of high molecular weight complexes (*Figures 4* and *5*). Negative stain electron microscopy experiments show an exponential distribution of complexes, with assemblies containing 2–4 stacked copies of LC8 clearly visualized in 2D projection averages and suggesting a high degree of flexibility within the duplex IDP linkers persists upon complex formation (*Figure 5*).

Multivalency and intrinsic disorder in ASCIZ's LC8 binding domain enable this dynamic ensemble of low occupancy complexes. Many proteins utilize multiple binding sites within intrinsically disordered regions for regulation, complex formation, and a multitude of other functions (*Cortese et al., 2008*; *Uversky, 2015*). In some examples, multiple binding sites serve as a scaffold to bring proteins together (*Cortese et al., 2008*), while in others, they modulate phase transitions that lead to the formation of bimolecular condensates (*Banani et al., 2017*; *Banani et al., 2016*). The diversity of these examples highlights the importance of multivalency and intrinsically disordered regions in protein function and regulation. It is of note that although ASCIZ's multiple binding sites are similar to those that lead to phase transitions in other systems (*Li et al., 2012*), we did not detect this behavior in

vitro. However, ASCIZ puncta formation has been observed in cell culture in response to treatment with MMS, a DNA methylation agent (*Jurado et al., 2012a*; *Jurado et al., 2010*), (*McNees et al., 2005*). These puncta do not form in the absence of LC8, indicating that LC8 binding to multiple recognition motifs is necessary for foci formation.

Comparison of ASCIZ with another LC8 multivalent binding partner, Nup159, underscores the uniqueness of the ASCIZ-LC8 assembly. While three other multivalent LC8 binding partners with more than two recognition motifs are known to exist (*Dunsch et al., 2012*; *Fejtova et al., 2009*; *Gupta et al., 2012*; *Stelter et al., 2007*), the role of multiple sites has only been characterized for ASCIZ (this work) and Nup159 (*Nyarko et al., 2013*). Nup159 cooperatively binds five LC8 dimers and forms a relatively stable complex readily visible by electron microscopy and 2D classification analysis (*Stelter et al., 2007*). As Nup159 has a slightly lower affinity for LC8 than ASCIZ (2.9 µM vs. 0.9 µM, respectively (*Nyarko et al., 2013*), the difference between the Nup159 grids with uniformly stacked structures and the ASCIZ grids with sparse and heterogeneous structures of 2–4 stacked LC8 dimers is intriguing. This difference in structural heterogeneity can be attributed to the higher flexibility or short life time of ASCIZ:LC8 complexes relative to Nup159:LC8 complexes, while the scarcity of high-occupancy states is consistent with a unique mode of negative cooperativity. Given the different function of Nup159 (in nuclear pore assembly) versus ASCIZ:LC8 complexes (transcription regulation), the data suggest that the dynamic properties and unique mechanism of assembly that is conserved in LC8-ASCIZ complexes may reflect an important feature required for autoregulation of LC8 transcription.

(3) ASCIZ regulates its transcriptional activity by binding multiple copies of its gene product, LC8. Cell culture based transcription assays demonstrate that ASCIZ affinity for LC8 is negatively correlated with transcriptional activity (*Figure 7*). Although the differences between each construct are very small, we see an obvious trend where fewer occupied LC8 binding motifs lead to increased ASCIZ transcriptional activity, and vice versa. The results of this assay suggest that LC8 concentration could fine-tune ASCIZ activity in a cellular environment by shifting the population of complexes towards higher or lower occupancy states.

A great example of multisite regulation is the E26 transformation-specific transcription factor (Ets-1), which tunes its transcriptional activity through multisite phosphorylation of its serine-rich domain. Phosphorylation of the serine rich region occludes the DNA-binding interface and stabilizes its helical inhibitory module, inhibiting Ets-1 DNA binding ~20 fold (*Desjardins et al., 2014*; *Lee et al., 2008*; *Pufall et al., 2005*). Similarly, the function of p53 is modulated by over 50 post-translational modifications that are proposed to be interdependent (*Meek and Anderson, 2009*). Phosphorylation of specific p53 residues prevents binding to the inhibitory protein HDM2, while increasing binding to the activating proteins CREB-binding protein (CBP) and p300 (*Ferreon et al., 2009*). p53 affinity for CBP/p300 depends on the extent of p53 phosphorylation; successive phosphorylation events increase p53 affinity for the TAZ1, TAZ2, and KIX domains of CBP/p300 (*Lee et al., 2010*; *Teufel et al., 2009*). p53 also binds to a multitude of other proteins that regulate its activity (*Beckerman and Prives, 2010*). While many other transcription factors tune their activity through multisite regulation, we could find no other examples besides ASCIZ where binding to multiple copies of their gene product modulates activity; yet, the prevalence of IDP domains in transcription factors indicates that such mechanisms are likely to be widespread and studies such as these are becoming more tractable with the integrated approaches used here.

## Model of ASCIZ transcriptional regulation

Based on our experimental data, we have developed a model of ASCIZ transcriptional regulation that illustrates the relationship between transcriptional activity (red arrow) and LC8 concentration (blue arrow) (*Figure 8*). As the cellular level of LC8 (blue dimers) increases, the number of LC8 molecules bound to ASCIZ also increases. The ASCIZ-LC8 complex primarily exists as a dynamic equilibrium of different complex stoichiometries and degrees of disorder (center brackets). We propose that this low occupancy conformational ensemble is important for maintaining a basal level of LC8 transcription. It acts as a buffer for changes in the concentration of LC8 and fine-tunes transcription levels according to cellular needs. A change in LC8 cellular concentration would shift this dynamic equilibrium to a higher or lower occupancy state without dramatically altering the level of transcription. All 11 (or 7) binding sites are therefore occupied in the heterogeneous mixture of complexes, as is demonstrated by NMR titration (*Figure 6*), and participate in maintaining a homeostatic

concentration of LC8. Thus, it is ensured that a high concentration of LC8 does not drastically decrease LC8 production, but rather shifts it to a lower level. As LC8 is a hub protein that interacts with >40 protein partners, and is predicted to bind to 100 additional proteins from a diverse selection of cellular pathways (*Rapali et al., 2011b*), maintaining a constant level of its transcription is essential.

# Materials and methods

Key resources table

| Reagent type (species) or resource | Designation | Source or reference | Identifiers | Additional information |
|---|---|---|---|---|
| Gene (*Homo sapiens*) | ASCIZ | NA | Uniprot ID: O43313 | |
| Gene (*Drosophila melanogaster*) | dASCIZ | NA | Uniprot ID: Q9VZU1 | |
| Cell line (*Mus musculus*) | ASCIZ knockout mouse embryonic fibroblast | PMID: 22167198 | | |
| Antibody | ASCIZ (rabbit monoclonal) | PMID: 15933716 | | WB: 100 ng/mL |
| Commercial assay or kit | Dual-luciferase reporter assay kit | Promega | Catalog number: E1910 | |
| Chemical compound, drug | D-glucose $^{13}C_6$ | Sigma Aldrich | Catalog number: 389374 | |
| Chemical compound, drug | Ammonium-$^{15}$N chloride | Sigma Aldrich | Catalog number: 299251 | |
| Software, algorithm | Origin 7.0 | OriginLab | | |
| Software, algorithm | SEDFIT | open-source | | |
| Software, algorithm | SEDNTERP | open-source | | |
| Software, algorithm | SEDPHAT | open-source | | |
| Software, algorithm | Curvefit | Palmer lab website | | |
| Software, algorithm | Topspin | Bruker Biospin Corporation | RRID:SCR_014227 | |
| Software, algorithm | Sparse Multidimentional Fourier Transform | Kozminski lab website | | |
| Software, algorithm | NMRFAM-Sparky | NMR facility at University of Wisconsin-Madison website | RRID:SCR_014228 | |
| Software, algorithm | ATSAS package | EMBL Hamburg BioSAXS website | RRID:SCR_015648 | |
| Software, algorithm | EMAN2 | National Center for Macromolecular Imaging | | |
| Software, algorithm | RELION 2.0 | open-source | | |

## Cloning, Protein Expression, and Purification

Studies were carried out using constructs from human ASCIZ (Uniprot O43313) as well as *Drosophila* ASCIZ (dASCIZ) (Uniprot Q9VZU1) which, with its fewer recognition motifs, smaller size, and available mutant phenotypes, is a tractable model of the human ASCIZ. Constructs of the dASCIZ zinc finger domain (ZnF) and the LC8 binding domain (dLBD) were generated by cloning residues 1–156 or 241–388, respectively, of *Drosophila* ASCIZ into the pET2Zt2-1a vector. The constructs were expressed in frame with a hexahistidine tag, Protein A solubility tag, and cleavage site for the tobacco etch virus (TEV) enzyme. Shorter constructs of the dLBD were generated by cloning residues 241–324 (QT1-3), 271–341 (QT2-4), 299–376 (QT4-6), and 321–388 (QT4-7) into the pET2Zt2-1a vector. The human LC8 binding domain (hLBD) construct was generated by cloning human ASCIZ (Uniprot O43313) residues 362–823 into the pET24d vector (Novagen) and expressing the construct in frame with a hexahistidine tag and TEV cleavage site.

For the five human ASCIZ mutants, ASCIZ AAA1-4, AAA8-11, AAA5-11, AAA1-4,8–11, and AAA-all, residues 7–9 of the LC8 binding motif (usually the residues TQT), were mutated to AAA to prevent binding. Mutations were performed using either the QuikChange Lightening Mutagenesis Kit

(Agilent) or by synthesizing short constructs (300–350 bp) containing the desired mutations and using Gibson Assembly (New England Biosciences, Ipswich) to insert them into the LC8-binding domain (hLBD) gene (residues 362–823 of human ASCIZ). All constructs were transformed into *Escherichia coli* Rosetta DE3 cells and expressed at 37°C in LB or minimal autoinduction media with $^{12}C$ or $^{13}C$ glycerol and $^{15}NH_4Cl$ as the sole carbon and nitrogen sources, respectively. Recombinant protein expression was induced with 0.4 mM IPTG (for LB cultures) and growth continued at 25°C for 16 hr. Cells were harvested and purified under denaturing conditions using TALON His-Tag Purification protocol (Clontech). The solubility tag and/or hexahistidine tag were cleaved by TEV protease and the protein was further purified using strong anion exchange chromatography (Bio-Rad, Hercules, California) followed by gel filtration on a Superdex$^{TM}$ 75 gel filtration column (GE Health). The purity of the recombinant proteins, as assessed by SDS-polyacrylamide gels, was >95%. The pure proteins were stored at 4°C and used within 1 week. LC8 was prepared as previously described (*Barbar et al., 2001*).

## Peptide design and synthesis
Peptides corresponding to the seven putative recognition sequences from dASCIZ were commercially synthesized: YMSSQKLDMETQTEE (QT1p), YLAPLLRDIETQTPD (QT2p), YTPDTRGDIGTMTDD (QT3p), DLQTSAHMYTQTCD (QT4p), EELGLSHIQTQTHW (QT5p), WPDGLYNTQHTQTCD (QT6p), and EPDNFQSTCTQTRW (QT7p) (GenScript, Piscataway, NJ). Non-native amino acids (underlined in *Table 2*) were added to the N-terminus to enhance solubility or concentration determination by UV absorbance at 280 nm.

## Isothermal Titration Calorimetry
Binding thermodynamics of the ASCIZ and dASCIZ construct/peptide-LC8 interactions were obtained at 25°C with a VP-ITC microcalorimeter (Microcal, Westborough, MA). The binding buffer was composed of 50 mM sodium phosphate, 50 mM sodium chloride, 1 mM sodium azide, 5 mM β-mercaptoethanol, pH 7.5. Protein concentrations were determined by absorbance measurement at 280 nm. Extinction coefficients for each construct are as follows. LC8 = 14,565 $M^{-1}cm^{-1}$, dLBD = 17,085 $M^{-1}cm^{-1}$, LBD = 15,470 $M^{-1}cm^{-1}$, QT1−3 = 2,980 $M^{-1}cm^{-1}$, QT2−4 = 2,980 $M^{-1}cm^{-1}$, QT4−6 = 10,095 $M^{-1}cm^{-1}$, QT4−7 = 14,105 $M^{-1}cm^{-1}$.

   dLBD was placed in the reaction cell at a concentration of 8 µM and titrated with LC8 at a concentration of 800 µM. For binding of dASCIZ constructs QT1-3, QT2-4, QT4-6, and QT4-7, 10 µM of construct was titrated with 400 µM LC8. For interactions with synthetic peptide, peptides were dissolved in binding buffer to a final concentration of 300 µM and then added to LC8 at a concentration of 30 µM in the reaction cell. ASCIZ hLBD and mutant hLBD constructs were placed in the reaction cell at a concentration of 9–16 µM and titrated with 900 µM LC8. Peak areas were integrated and data were fit to a single-site binding model in Origin 7.0 from which the stoichiometry (*N*), dissociation constant (*K_d*), and the change in enthalpy (Δ*H*), and entropy (Δ*S*) were obtained. Reported data are the average of two or more independent experiments. As the binding-model fit was very good and data were reproducible, error was determined based on a 5% uncertainty in protein concentration calculations.

## Circular dichroism
CD experiments were conducted on a Jasco720 spectropolarimeter in a 1 mm cell. For the spectrum of dLBD and smaller constructs, ten scans were averaged at a concentration of 30 µM in a buffer composed of 10 mM sodium phosphate, pH 7.5, at 25°C and 10°C. For the ZnF, ten scans were averaged at a concentration of 25 µM in a buffer composed of 10 mM sodium phosphate, 200 mM sodium sulfate, 50 µM zinc sulfate, pH 7.5, at 10°C, 25°C, and 35°C.

## Analytical ultracentrifugation
Sedimentation velocity experiments for the titration of dLBD and LC8 were performed in a Beckman Coulter Model XL-I analytical ultracentrifuge equipped with UV/Vis scanning optics. Reference (400 µL binding buffer; 50 mM sodium phosphate, 50 mM sodium chloride, 1 mM sodium azide, 5 mM TCEP, pH 7.5) and sample (380 µL) solutions were loaded into 12 mm double-sector cells with quartz windows and the cells were then mounted in an An-50 Ti 8-hole rotor. LC8 was prepared at a

concentration of 15 µM while the concentration of dLBD was varied from 15 to 1.5 µM. Proteins were centrifuged at 50,000 rpm at 20°C, and radial absorbance data were collected at appropriate wavelengths in continuous mode every 5 min without averaging. Data were fit to a continuous size-distribution [c(S)] model using the program SEDFIT (*Schuck, 2000*). The partial specific volume of the proteins, buffer density, and buffer viscosity were computed using the program SEDNTERP (*Hayes and Philo, 1995*).

Sedimentation velocity experiments for the ZnF domain were performed on a Beckman Proteo-meLab XL-A/XL-I analytical ultracentrifuge in a buffer composed of 50 mM sodium phosphate, 200 mM sodium chloride, 0.4 mM zinc sulfate, 1 mM sodium azide, 2 mM TCEP, pH 7.0. The sample was centrifuged at 40,000 rpm at 20°C for 7 hr and absorbance data were collected at 286 nm. Data were fit to a continuous size-distribution [c(s)] model using the program SEDPHAT (*Vistica et al., 2004*).

## NMR experiments

NMR measurements were collected at 10°C, using 300–350 µM isotopically ($^{13}$C/$^{15}$N or $^{15}$N) labeled dLBD in a buffer at pH 6.5 composed of 10 mM sodium phosphate, 10 mM sodium chloride, 1 mM sodium azide, 10 mM β-mercaptoethanol, a protease inhibitor mixture (Roche Applied Science, Madison, WI), and 2–2 dimethylsilapentane-5-sulfonic acid for $^1$H chemical shifts referencing. Data for backbone assignments were collected on a Bruker Avance 850 MHz spectrometer equipped with a cryoprobe. Five-dimensional HN(CA)CONH and HabCabCONH experiments (*Kazimierczuk et al., 2010*; *Motáčková et al., 2010*; *Nováček et al., 2011*) and a three-dimensional HNCO experiment were acquired with non-uniform sampling of the indirectly detected dimensions and used for sequential assignment of $^{13}$C-$^{15}$N- dLBD.

Interaction of unlabeled LC8 and $^{13}$C-$^{15}$N labeled dLBD was characterized by collecting three-dimensional BEST-TROSY-HNCO spectra at multiple molar ratios of LC8, 1: 0.25 (dLBD: LC8), 1:1, 1:2, 1:5, and 1:8. For the interaction of unlabeled LC8 with $^{15}$N-labeled QT2-4 (residues 271–341) or QT4-6 (residues 321–376), two-dimensional BEST-TROSY-HSQC spectra were collected at the molar ratios (QT2-4/QT4-6:LC8) 1:0.25, 1:1, 1:2, 1:3, and 1:4. NMR titration data were analyzed and plotted by measuring peak volumes using Sparky and averaging over each 10 amino acid QT motif.

HNCO-based $R_1$ relaxation measurements experiments were recorded with relaxation delay times ranging from 11.2 to 2352 ms, and the $R_2$ relaxation data were acquired using relaxation delays ranging from 14.4 to 259 ms. Sixteen total $R_1$ or $R_2$ experiments were recorded, including six duplicate experiments for error determination. Curve fitting was performed using the rate analysis script Sparky2Rate and the program Curvefit (A. G. Palmer, Columbia University). Steady-state $^1$H−$^{15}$N heteronuclear NOEs were acquired using 6 s total saturation time. Error bars were determined from the intensities of the baseline noise using the formula $\sigma/(NOE) = [(\sigma I_{sat}/I_{sat})^2 + (\sigma I_{unsat}/I_{unsat})^2]^{1/2}$, where $I_{sat}$ and $\sigma I_{sat}$ correspond to the intensity of the peak and its baseline noise.

All two-dimensional spectra and the three-dimensional HNCO spectra were processed using Top-Spin (Bruker Biosciences; RRID:SCR_014227), and the non- uniformly sampled five-dimensional HN (CA)CONH and HabCabCONH spectra were processed with Sparse Multidimensional Fourier Transform (*Kazimierczuk et al., 2009*; *Stanek and Koźmiński, 2010*), (the software for data processing is available online at the Warsaw University Laboratory (nmr.cent3.uw.edu.pl/software)). All spectra were analyzed with the graphical NMR assignment and integration software NMRFAM-Sparky (RRID:SCR_014228).

## Analytical Size Exclusion Chromatography and Native Gel Titration

hLBD at a concentration of 30 µM was incubated with 600 µM LC8 at various molar ratios: (hLBD: LC8) 1:3, 1:5, 1:7, 1:9, 1:11, and 1:13. The complex was loaded on a Superdex 200 analytical column (GE healthcare, Wauwatosa, WI) in binding buffer: 50 mM sodium phosphate, 50 mM NaCl, 5 mM β-mercaptoethanol, 1 mM sodium azide, pH 7.5. 100 or 200 µl of protein samples were injected at a flow rate of 0.5 ml/min at room temperature and samples were monitored by UV absorption at 280 nm.

For native gel electrophoresis titrations, dLBD or hLBD and LC8 were incubated at the molar ratios listed above and run on a 10% polyacrylamide gel at a constant 10 mAmps for 5–7 hr.

## Small Angle X-ray Scattering

Small-angle X-ray scattering experiments were conducted at the ESRF BioSAXS beamline BM29 (*Pernot et al., 2013*) in Grenoble, France. dLBD and LC8 samples were purified as described above and dialyzed into binding buffer (50 mM sodium phosphate, 50 mM sodium chloride, 10 mM beta-mercaptoethanol, 1 mM sodium azide, pH 7.5) before SAXS measurements. 30 µl of dLBD:LC8 complex (1:8 molar ratio) at five different concentrations for each sample (and buffer) were exposed to X-rays and scattering data collected using the robotic sample handling available at the beamline. 10 individual frames were collected for every exposure, each 2 s in duration using the Pilatus 1M detector (Dectris, Switzerland). Individual frames were processed automatically and independently within the EDNA framework, yielding individual radially averaged curves of normalized intensity versus scattering angle $s = 4\pi Sin\theta/\lambda$. Additional data reduction within EDNA utilizes the automatic data processing tools of EMBL-Hamburg ATSAS package (M. V. K. Petoukhov, P. V.; Kikhney A. G.; Svergun D. I., 2007), to combine timeframes, excluding any data points affected by aggregation induced by radiation damage, yielding the average scattering curve for each exposure series. Matched buffer measurements taken before and after every sample were averaged and used for background subtraction. Merging of separate concentrations and further analysis steps were performed manually using the tools of the ATSAS package (M. V. K. Petoukhov, P. V.; Kikhney A. G.; Svergun D. I., 2007) (RRID:SCR_015648). The forward scattering I(0) radius of gyration, Rg were calculated from the Guinier approximation (A., 1938), the hydrated particle volume was computed using the Porod invariant (*Porod, 1982*) and the maximum particle size $D_{max}$, was determined from the pair distribution function computed by GNOM (*Svergun, 1992*) using PRIMUS.

## Electron Microscopy

Electron microscopy (EM) studies were conducted using dLBD and hLBD incubated with a molar excess of LC8, and the formed complexes were negatively stained for contrast enhancement using established protocols (*Myers et al., 2017*). Briefly, dLBD (50 nM) was mixed with LC8 at a molar ratio 1:8, and human hLBD peptide (50 nm) was mixed with LC8 at a molar ratio of 1:13, in EM buffer containing 20 mM Tris, pH 7.5, 50 mM NaCl, 10 mM BME and 1 mM NaN$_3$. A 3 µl drop of sample was applied to a glow-discharged continuous carbon coated EM specimen grid (400 mesh Cu grid, Ted Pella, Redding, CA). Excess protein was removed by blotting with filter paper and washing the grid two times with EM buffer. The specimen was then stained with freshly prepared 0.75% (wt vol$^{-1}$) uranyl formate (SPI-Chem, West Chester, PA).

Negatively stained specimens were visualized on a 120 kV TEM (iCorr, FEI, Hillsboro, OR) at a nominal magnification of 49,000x at the specimen level. Digital micrographs were recorded on a 2K × 2K CCD camera (FEI Eagle) with a calibrated pixel size of 4.37 Å pixel$^{-1}$ and a defocus of 2.0–3.5 µm. For the dLBD-LC8 specimen, a total of 2574 single particle images were extracted from ~300 micrographs, and for hLBD-LC8, 1234 particles were extracted from ~200 micrographs. Complexes with clear oligomeric structure could be identified and were manually-selected using EMAN2 (*Tang et al., 2007*). Single particle images were extracted with a box size of 160 × 160 pixels and CTF-corrected (phase-flipped) in EMAN2. Reference-free 2D class averages were generated in EMAN2 and RELION 2.0 (*Scheres, 2012*) using CTF-corrected and high-pass filtered image datasets. Statistical analysis of oligomeric composition was performed by counting the number of subunits identified from single particle images and classifying them manually as 2 – 7mers (dLBD:LC8 complexes) or 2 – 11mers (hLBD:LC8 complexes) (*Figure 5—figure supplement 1*). Particles that could not be confidently assigned were discarded, leaving 2334 oligomers assigned for the dLBD: LC8 and 967 for hLBD:LC8 datasets. Complexes containing only a single LC8 dimer could not be distinguished from unbound LC8 particles, and were not included in our analysis.

As a positive control, Nucleoporin159 (Nup159) in complex with LC8 was also prepared for negative stain EM under similar conditions to the dLBD/hLBD samples, and as previously described (*Stelter et al., 2007*) (not shown). As a negative control, we prepared EM grids with LC8 alone and dLBD/hLBD alone. No oligomeric structures (*i.e.* beads on a string) were observed in these images (not shown).

## Transcription reporter assays

To measure transcriptional activity of ASCIZ mutants, six ASCIZ constructs were cloned into the pEGFP vector (Clontech): WT ASCIZ (1-823), ΔZnF (230-823), ASCIZ AAA1-4, ASCIZ AAA8-11, ASCIZ AAA5-11, and ASCIZ AAA1-4, 8–11. Approximately 2 kbp of the Dynll1 promoter was cloned into the pGL3 vector (Promega, Madison, WI) upstream of the firefly luciferase gene as previously described (*Jurado et al., 2012a*). Using FuGENE 6 (Promega), immortalized ASCIZ knockout mouse embryonic fibroblasts (MEFs) (*Jurado et al., 2010*) were co-transfected with ASCIZ constructs, the Dynll1 promoter, and a pRL-CMV vector containing *Renilla* luciferase for normalization of firefly/luciferase ratios. The authenticity of the ASCIZ knockout MEF cell line was verified by PCR genotyping of the ASCIZ locus. The cell line tested to be free from mycoplasma contamination using a MycoAlert mycoplasma detection kit (Lonza, Switzerland). 24 hr after transfection, cells were transferred to 96-well plates and incubated overnight before determining reporter gene activities using the dual-luciferase reporter assay kit (Promega) and a Polarstar Optima (BMG Labtechnologies, Germany) instrument. For assessment of protein expression levels, human U2OS cells were transfected with ASCIZ constructs using FuGENE six and were probed with ASCIZ antibody (*McNees et al., 2005*).

## BMRB accession code

The chemical shifts for dLBD ASCIZ have been deposited in the Biological Magnetic Resonance Data Bank under accession code 27412.

## Acknowledgements

The authors are indebted to Professor Clare Woodward for the many helpful discussions. This work was supported by National Institutes of Health Grant GM 084276 to EB and by an Impact Award from the College of Science at Oregon State University. Support to facilities includes the Oregon State University NMR Facility funded in part by the National Institutes of Health, HEI Grant 1S10OD018518, and by the M. J. Murdock Charitable Trust grant # 2014162. Access to the NMR facility of CEITEC Masaryk University was provided by iNEXT, project number 653706, funded by the Horizon 2020 programme of the European Union. This article reflects only the author's view and the European Commission is not responsible for any use that may be made of the information it contains. JH was supported by the National Health and Medical Research Council of Australia (Senior Research Fellowship APP1022469 and Project Grant APP1026125) and Victorian State Government Operational Infrastructure Support. Electron microscopy was conducted with support from the Multiscale Microscopy Core (OHSU), Advanced Computing Center (OHSU) and the National Institutes of Health Grant R35GM124779 to SLR. Small angle X-ray scattering data were collected at the European Molecular Biology Lab in Grenoble, France.

## Additional information

### Funding

| Funder | Grant reference number | Author |
| --- | --- | --- |
| National Health and Medical Research Council | APP1026125 | Jörg Heierhorst |
| National Health and Medical Research Council | APP1022469 | Jörg Heierhorst |
| Victorian State Government | Operational Infrastructure Support | Jörg Heierhorst |
| National Institute of General Medical Sciences | R35GM124779 | Steve L Reichow |
| National Institute of General Medical Sciences | R01-084276 | Elisar J Barbar |
| College of Science at Oregon State University | Impact Award | Elisar J Barbar |

The funders had no role in study design, data collection and interpretation, or the decision to submit the work for publication.

## Author contributions
Sarah Clark, Conceptualization, Formal analysis, Validation, Investigation, Visualization, Writing—original draft, Writing—review and editing; Janette B Myers, Formal analysis, Methodology; Ashleigh King, Validation, Methodology; Radovan Fiala, Resources, Methodology, Writing—review and editing; Jiri Novacek, Resources, Software, Methodology; Grant Pearce, Resources, Formal analysis, Funding acquisition, Methodology, Writing—review and editing; Jörg Heierhorst, Supervision, Funding acquisition, Writing—original draft, Writing—review and editing; Steve L Reichow, Formal analysis, Supervision, Funding acquisition, Visualization, Methodology, Writing—original draft, Writing—review and editing; Elisar J Barbar, Conceptualization, Resources, Formal analysis, Supervision, Funding acquisition, Validation, Investigation, Methodology, Writing—original draft, Project administration, Writing—review and editing

## Author ORCIDs
Janette B Myers (iD) http://orcid.org/0000-0002-7758-2649
Grant Pearce (iD) http://orcid.org/0000-0002-2683-0331
Jörg Heierhorst (iD) http://orcid.org/0000-0003-2789-9514
Elisar J Barbar (iD) http://orcid.org/0000-0003-4892-5259

## Decision letter and Author response
Decision letter https://doi.org/10.7554/eLife.36258.027
Author response https://doi.org/10.7554/eLife.36258.028

# Additional files

## Supplementary files
• Transparent reporting form
DOI: https://doi.org/10.7554/eLife.36258.023

## Data availability
The chemical shifts for dLBD ASCIZ have been deposited in the Biological Magnetic Resonance Data Bank under accession code 27412 (http://www.bmrb.wisc.edu/data_library/summary/index.php?bmrbId=27412).

The following dataset was generated:

| Author(s) | Year | Dataset title | Dataset URL | Database, license, and accessibility information |
|---|---|---|---|---|
| Sarah C, Elisar B, Radovan F, Jiri N | 2018 | dASCIZ LC8 binding domain (residues 241-388) | http://www.bmrb.wisc.edu/data_library/summary/index.php?bmrbId=27412 | Publicly available at the Biological Magnetic Resonance Data Bank (accession no. 27412) |

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
