## [Decision Letter]

[Editors’ note: a previous version of this study was rejected after peer review, but the authors submitted for reconsideration. The first decision letter after peer review is shown below.]

Thank you for submitting your work entitled "Multivalency in Transcriptional Regulation of the Hub Protein LC8" for consideration by *eLife*. Your article has been evaluated by a Senior Editor and three reviewers, one of whom is a member of our Board of Reviewing Editors. The reviewers have opted to remain anonymous.

Our decision has been reached after consultation between the reviewers. Based on these discussions and the individual reviews below, we regret to inform you that your work will not be considered further for publication in *eLife*.

"Multivalency in Transcriptional Regulation of the Hub Protein LC8" describes a biophysical study that focuses on the mechanism of transcriptional control of the dynein light chain (LC8) through binding interactions with the disordered region of the transcription factor ASCIZ. All three reviewers felt that this is a fascinating system and that the work does help in understanding of how regulation does occur. Yet there were some reservations about the novelty, with significantly more experiments required at the in-vitro and cellular level to prove the hypothesis regarding the feedback mechanism regulating expression. In addition, it was felt that the work would be strengthened by additional NMR experiments to more fully characterize the binding. Thus, although all reviewers felt that the work was of interest, all noted weaknesses that prohibit publication in *eLife*.

*Reviewer #1:*

This work describes an elegant biophysical study that helps elucidate the mechanism of transcriptional control of the LC8 protein that is implicated in a wide range of biological processes. The authors come up with a mechanism whereby the disordered domain of the transcription factor ASCIZ is able to bind multiple copies of LC8, in effect, sensing its concentration and controlling levels of transcription of the LC8 gene through a competition with activator binding. Overall this is an exciting development, but I do believe that the work could be clarified by consideration of a number of points below.

1) The authors show by NMR that as a function of titration signal is lost due to exchange and increase in molecular weight. The major insight by NMR is that there are two different classes of binding regions that bind either 3 or 4 LC8 dimers with different affinities. Yet little mechanism into binding order to sites within these classes is obtained and as the authors mention the affinities are very different from ITC. Can the authors look at say smaller constructs that contain only 3 or only the 4 dimer sites in an effort to circumvent the issue with disappearing peaks? Indeed even smaller constructs would allow to address whether at low stoichiometries there is preferential binding to one of the sites, whether there is exchange between sites and what both the kinetics and thermodynamics might be.

2) Related to 1, I find it interesting and surprising that upon saturating binding there is a 1:3 complex that appears to be dominant (can the authors give the relative amounts of 1:3 vs. 1:7). I don't understand why the 1:3 can’t be saturated (pushed to fully bound) – simply thermodynamics would dictate this unless there is some sort of structural change. What is the mechanism involved here? And what are the implications for transcription, since even at very high concentrations of LC8 it appears that there will be some activation, which seems wasteful. The authors need to discuss this point more fully in terms of what the biological ramifications are/mean.

3) Related to 2 can the authors rationalize some of their transcription results where removing a number of ASCIZ sites seems not to matter?

4) In the proposed model there is a duplex of dLBD that is formed that aids in LC8 binding. Perhaps I am missing something, but the duplex evidence comes from the centrifugation results (or are there additional data?). How conclusive is the centrifuge results and if these are the primary data then please expand the Discussion.

5) More discussion should be given to other transcription systems. There is beautiful NMR work by McIntosh on ETS that shows a gradual change in activity with phosphorylation that is not discussed. It should be.

*Reviewer #2:*

The manuscript by Clark, et al., reports data on the binding of dynein light chain (LC8) to the disordered C-terminal region (CTR) of the zinc finger transcription factor ASCIZ. Others have reported that ASCIZ regulates the expression of LC8 in certain cell types, and in some cancer cell lines, and that disruption of this regulatory circuit is associated with mitotic defects seen previously with loss of LC8. In addition, others have reported that the CTR of ASCIZ contains multiple short linear motifs (SLIMs) with the sequence TQT known to bind to LC8. Here, the authors characterized seven LC8 binding SLIMs within the CTR of *Drosophila* ASCIZ using ITC, AUC, and NMR. As LC8 was titrated into the ASCIZ CTR, a mixture of sub-stoichiometric and 1:7 complexes were formed, presumably with two CTRs stitched together by multiple LC8 dimers (as proposed by the authors). Mutagenesis of several different combinations of multiple SLIMs confirmed their LC8 binding properties and allowed the rank order of their μM affinities to be determined. The authors noted that the affinity of the CTR, with multiple binding sites, for LC8 was higher than that of isolated binding site, but this point was not rigorously addressed in the manuscript. The authors extended their studies to the CTR of human ASCIZ, which they showed contains 11 LC8 binding SLIMs using ITC and mutagenesis. Finally, the authors performed a Luc-reporter transcription assay using wt and mutant ASCIZ expressed in ASCIZ null MEFs, which showed that mutation of LC8 binding SLIMs was associated with a slight increase in expression of the luc reporter. This latter finding suggests that LC8 exerts negative feedback on its expression by binding to the SLIMs within the CTR of ASCIZ, possibly altering DNA binding and/or interactions with other transcription factors. While the authors' hypothesis regarding this negative feedback mechanism is novel and interesting, the data in the manuscript fall well short of proving the hypothesis. Much more extensive in vitro and in cell data would be needed to establish the LC8 concentration dependence of the negative feedback circuit and further data required to elucidate the associated mechanism of feedback. In addition, the new knowledge gained regarding the identification of LC8 binding SLIMs within fly and human ASCIZ seems incremental with respect to the 2011 BBRC report from Rapali, et al. Therefore, the manuscript does not provide the level of rigor, novelty and impact associated with *eLife* but rather would be more suitable for a more technical journal.

*Reviewer #3:*

Summary:

This manuscript describes the characterization of the disordered C-terminal region (LBD) of the transcription factor ASCIZ and its interaction via multiple short linear motifs to LC8. The binding regulates ASCIZ activity and provides a feedback loop for controlling transcription of LC8. Different biophysical tools are used to study the dynamic complex between the ASCIZ LBD and LC8, including ITC, analytical ultracentrifugation and NMR on primarily dASCIZ constructs but also including some human. Results clearly support a model with a dynamic (based on NMR) mixture of complexes with different LC8 occupancies (based on centrifugation) and with ASCIZ engaging LC8 at multiple different motifs. Arguments are made for the cooperativity of multivalent binding (based on ITC) and for the ability of this dynamic mixture of states enabling tunable transcription levels (based on transcription assays).

General assessment:

Overall the manuscript describes extensive work on a fascinating system that offers insights into a mechanism for tunable regulation of transcription factor activity. The technical work appears to have been performed at a high level. The results are important for demonstrating a powerful example of the regulatory effects of multivalent binding of motifs within intrinsically disordered protein regions, one that could be more general. However, the authors do not place this example within the context of other literature on dynamic complexes involving multiple motifs of disordered proteins or other mechanistic approaches for generating tunable activity in transcription factors.

1) The authors should acknowledge this as an example of a dynamic or fuzzy complex and cite reviews (M Fuxreiter or others). The novelty of this being a unique case of multisite recognition is somewhat overstated. The manuscript should cite examples of other proteins that interact with multiple motifs and compare mechanistic effects of multi-site recognition with this mechanism. For two sites, this could include the p53 transactivation subdomains that bind the CBP TAZ2 (P Wright) and for larger numbers similar to that studied here it could include 14-3-3 targets such as Raf-1 protein kinase and yeast Cdc25 (multiple authors), CFTR regulatory domain (J Forman-Kay), Sic1 (M Tyers) and even phase separation via multiple motifs and multiple modular binding domains (M Rosen). Examining the approaches used to quantify local interactions or the caveats described in these cases from the literature may also be valuable for interpretation of the LC8-LBD interactions.

2) It is equally important to cite examples of transcription factors that are regulated in a tunable manner such as Ets^-1^ (L McIntosh), as not all are binary on/off and probably many are tunable with mechanisms yet to be described but likely involving intrinsically disordered regions. The novelty of this being the only transcription factor to not be regulated in a binary manner is thus overstated.

3) "we consider a decrease in peak intensity as a measure for an increase in complex formation" and "It should be noted that these Kd values measured by NMR titration differ from those measured by ITC because the loss in peak intensity detectable by NMR is influenced by a number of factors in addition to direct LC8 interaction. These include exchange with neighboring sites, change in correlation time due to dimerization of the dLBD upon binding dimeric LC8, and increased correlation time as LC8 sites are saturated. These Kd values, therefore, identify different behavior of each set of motifs, but do not accurately determine specific motif binding constants."

Generally, different Kd values are expected for global (ITC) and local affinities so the local measures should not match the global, even if the local affinities are accurate. The authors should try to calculate a global Kd based on linked local affinities. Importantly, however, the local affinities are not likely to be accurate. In addition to the caveats mentioned by the authors, there is a strong possibility of intramolecular interactions leading to a lower intensity in the free LBD with binding to LC8 disrupting these intramolecular interactions and leading to increase of intensity for some LBD regions. This should be acknowledged and is especially important given the suggestion of "slightly more ordered structure in its C-terminal half" of the LBD. (Note that "ordered" would be better described as "transient", given the overall disordered nature of the LBD.) The differing apparent correlation times for LBD residues depending on distances from direct LC8 binding site(s) on the LBD and from the termini should also be acknowledged (P Wright and J Dyson). The complexity of interpretation of intensities needs to be laid out even if trying to make a simple interpretation. Ultimately, this reviewer is not convinced of the validity of the "two sub-domains" identified based on these different Kd and cooperativity values, due to the lack of ability to quantify local affinities from the NMR intensities.

4) Some background information on the nature of LC8 structure and its dimerization interface would be valuable. In particular, how does LC8 dimerization affect the cooperativity of binding? Can numerical models be generated to estimate the effect of dimerization in the context of linked multivalent interactions?

5) It would be valuable in the Discussion to at least briefly address the numbers of LC8-binding motifs found in known binding partners and how these proteins may exploit multiple motifs similarly (or not) to ASCIZ (based on: "Although some LC8 partners have multiple recognition motifs (Gupta, Diener, Sivadas, Rosenbaum, and Yang, 2012; Jie, Lohr, and Barbar, 2015; Moutin et al., 2014; Nyarko, Song, Novacek, Zidek, and Barbar, 2013; Slevin, Romes, Dandulakis, and Slep, 2014), human ASCIZ contains 11 functional LC8 binding sites (Rapali, Garcia-Mayoral, et al., 2011), the most of any partner protein.")

6) What are the estimated cellular concentrations of LC8 at various stages and how would these affect the mixture of complexes between ASCIZ and LC8? What might be the dominant complex of LC8-human ASCIZ in cells based on the binding parameters determined in this manuscript? What would be the minimum number of LC8 to inhibit transcription? How is the affinity or kinetics comparable to the transcriptional activator?

7) Given the high level of multivalency in the system with 11 binding motifs and a dimeric binding domain (see work of M Rosen), it seems possible that puncta or large-scale associated states may be observed in vitro or in cells. Is there any evidence for this?

[Editors’ note: what now follows is the decision letter after the authors submitted for further consideration.]

Thank you for submitting your revised manuscript "Multivalency regulates activity in an intrinsically disordered transcription factor" to *eLife* for consideration. Your article was reviewed by the same reviewers as for the previous submission.

All three reviewers were impressed with the amount of new biophysical data added to the paper that considerably improved the characterization of this complex system. It was also felt that the literature was reviewed much more thoroughly this time and the current work framed in a better context. The consensus was that the biophysical characterization of a difficult system was strong, while the in-vitro and cellular experiments to prove the hypothesis regarding the feedback mechanism regulating expression were weaker. At issue is that while the overall trend regarding binding and activity is consistent with your claims, the effect is small (Figure 7D) and the errors are large, and in some cases it appears that the differences in the points corresponding to different Kd's are not statistically significant. It was the consensus of the reviewers that a revision should focus more on the biophysics, while suggesting biological relevance through the presentation of the transcription data. If possible, additional transcriptional experiments that attempt to decrease errors to strengthen this section would be recommended but are not required. Please provide a detailed point-by-point response to each of the issues raised below to assist the reviewers in an evaluation of your revised manuscript.

1) As mentioned above the authors present an interesting and extensive biophysical study that characterizes the negative regulation of a transcription factor through binding of a main target protein. The reviewers feel that this (the biophysics) should be emphasized more strongly in a revised manuscript. It would strengthen the paper considerably if the errors associated with the transcription data of Figure 7 could be reduced, but this will be left at the discretion of the authors.

2) In the third paragraph of the subsection “Structure and Distribution of LBD:LC8 Complexes Visualized by Single Particle EM” the authors state that high order oligomers are rare with fully formed complexes almost never observed. Yet in their model of Figure 8 they propose that under high concentrations of LC8 a fully formed complex is formed. There seems to be a discrepancy here. Further it is hard for this reviewer to understand the role of negative cooperativity. If in fact the complete binding range is not used (i.e., one does not get fully bound transcription factor) then it seems like the full dynamic range over which changes in LC8 concentration occur is not optimally sampled by ASCIZ. What is the biological significance of the negative cooperativity, such that the full number of binding sites are not utilized? This seems not to be a very sensitive way of monitoring LC8 concentration.

---

## [Author Response]

[Editors’ note: the author responses to the first round of peer review follow.]

"Multivalency in Transcriptional Regulation of the Hub Protein LC8" describes a biophysical study that focuses on the mechanism of transcriptional control of the dynein light chain (LC8) through binding interactions with the disordered region of the transcription factor ASCIZ. All three reviewers felt that this is a fascinating system and that the work does help in understanding of how regulation does occur. Yet there were some reservations about the novelty, with significantly more experiments required at the in-vitro and cellular level to prove the hypothesis regarding the feedback mechanism regulating expression. In addition, it was felt that the work would be strengthened by additional NMR experiments to more fully characterize the binding. Thus, although all reviewers felt that the work was of interest, all noted weaknesses that prohibit publication in eLife.

In the previous review, dated July 31^st^, 2017, all three reviewers agreed that this is a fascinating system of high impact. Yet there was some reservation about publication in *eLife* in its current form and reviewers suggested several ways to strengthen the work. In this revision we address all the reviewers concerns and add substantial amount of data that considerably strengthen our model. Briefly, the new data include negative stain electron microscopy that give a structural view and quantitative distribution of the multiple disordered complexes in dynamic equilibrium (Figure 5 and Figure 5—figure supplement 1). Negative stain electron microscopy is particularly challenging for disordered systems and therefore the work here pushes the limit of this technique. New data for characterizing the dynamic intermediate were also obtained using small angle X-ray scattering and gel filtration (Figure 4 and Figure 4—figure supplement 1). We have collected significantly more NMR data (Figure 6 and Figure 6—figure supplement 1) and changed the Introduction (Figure 1) and interpretation of the model (Figure 8). As a result, this manuscript is much more focused and we hope is now suitable for publication in *eLife*. Below we briefly address the reviewers’ major concerns.

Reviewer #1:[…] 1) The authors show by NMR that as a function of titration signal is lost due to exchange and increase in molecular weight. The major insight by NMR is that there are two different classes of binding regions that bind either 3 or 4 LC8 dimers with different affinities. Yet little mechanism into binding order to sites within these classes is obtained and as the authors mention the affinities are very different from ITC. Can the authors look at say smaller constructs that contain only 3 or only the 4 dimer sites in an effort to circumvent the issue with disappearing peaks? Indeed even smaller constructs would allow to address whether at low stoichiometries there is preferential binding to one of the sites, whether there is exchange between sites and what both the kinetics and thermodynamics might be.

NMR titration data of two smaller constructs that contain only 3 binding sites are now included (Figure 6B, C). NMR titration of the smaller constructs substantiates our data from the full-length construct, which demonstrate that the C-terminal sites bind first. Excerpts from the manuscript are below.

“Plots of the average peak intensity (I/I_0_) for each 10-amino acid motif in dLBD, and in each of QT2-4 and QT4-6 constructs clearly show the dichotomy in the pattern of peak attenuation (Figure 6D-G). […] NMR dynamics experiments also demonstrate that residues in the C-terminal motifs are slightly more ordered in comparison to the N-terminal motifs (Figure 2F-G, Figure 2—figure supplement 1A), which may explain the tighter LC8 binding to this region.”

2) Related to 1, I find it interesting and surprising that upon saturating binding there is a 1:3 complex that appears to be dominant (can the authors give the relative amounts of 1:3 vs. 1:7). I don't understand why the 1:3 can’t be saturated (pushed to fully bound) – simply thermodynamics would dictate this unless there is some sort of structural change. What is the mechanism involved here? And what are the implications for transcription, since even at very high concentrations of LC8 it appears that there will be some activation, which seems wasteful. The authors need to discuss this point more fully in terms of what the biological ramifications are/mean.

We agree with the reviewer that this binding behavior is unusual and surprising. Because of this we have performed every experiment conceivable to verify and rationalize this apparent discrepancy between ITC and AUC. While high occupancy complexes are evident in AUC profiles, consistent with the ITC results, these complexes are in equilibrium with many smaller sub-saturated species, the most highly populated of which is a mixture of 1:2-1:4 complexes of dLBD:LC8. The low occupancy complexes are favored, relative to higher occupancy complexes, even in samples having a large excess of LC8. The presence of stable, low occupancy complexes is supported by new small angle X-ray scattering (SAXS) data, negative stain electron microscopy, and gel filtration. Importantly, also new to this submission, we show a similar behavior for human ASCIZ, and discuss that negative cooperativity in the transition from low occupancy to high occupancy complexes is a conserved feature of ASCIZ-LC8 interactions and explains the higher population of the low occupancy complexes. The completely revised discussion speculates on the biological ramifications of the ability of ASCIZ to undergo both positive and negative cooperativity.

3) Related to 2 can the authors rationalize some of their transcription results where removing a number of ASCIZ sites seems not to matter?

In the AAA 8-11 construct, removal of 4 sites decreases transcriptional activity, rather than increasing it as one would expect. We now state in the text that this is likely due to the higher affinity of the AAA 8-11 construct for LC8 relative to WT. All of the other AAA mutants show a decrease in binding affinity that is correlated with an increase in transcriptional activity. Therefore, despite the unexpected result where removing LC8 sites decreases activity, it in fact fits the trend perfectly (see Figure 7D, a new figure in this submission).

4) In the proposed model there is a duplex of dLBD that is formed that aids in LC8 binding. Perhaps I am missing something, but the duplex evidence comes from the centrifugation results (or are there additional data?). How conclusive is the centrifuge results and if these are the primary data then please expand the Discussion.

Evidence for duplex formation comes from a 17+ years of research by the authors on LC8 and its partners, summarized now in the Introduction and Figure 1. Also, in this work, several pieces of data support this interpretation: molecular weight of the complex from AUC, binding stoichiometry from ITC, and negative stain electron microscopy images. The negative stain electron microscopy data are new to the revised manuscript and provide a clear image for duplex formation.

5) More discussion should be given to other transcription systems. There is beautiful NMR work by McIntosh on ETS that shows a gradual change in activity with phosphorylation that is not discussed. It should be.Reviewer #2:[…] Much more extensive in vitro and in cell data would be needed to establish the LC8 concentration dependence of the negative feedback circuit and further data required to elucidate the associated mechanism of feedback.

We realize that our original hypothesis was not fully supported by the available data. The original hypothesis has been altered and now states “ASCIZ and LC8 engage in a dynamic equilibrium of multivalent interactions that tune the level of ASCIZ transcriptional activity.” Additional data is included in this manuscript to support our hypothesis. Using a combination of methods that include SAXS, AUC, gel filtration, native gel, and negative stain electron microscopy, we demonstrate that both the human and *Drosophila* proteins form a dynamic complex with LC8. Our cell culture transcription assays show a gradient of ASCIZ activity that depends on the number of LC8 dimers bound. We speculate that this dynamic complex is important for fine tuning ASCIZ transcriptional activity.

This work will be of particular relevance to the NMR, intrinsically disordered proteins and transcription regulation communities as it is transformative on several fronts: 1) it presents an attractive model that begins to explain how the concentration of an essential hub protein is maintained in the cell, 2) it gives a unique demonstration of how protein disorder and multivalency, conserved among species, regulate transcription, 3) it presents an example of a multivalent protein that undergoes both positive and negative cooperativity where binding of short peptides and longer constructs is quite different from binding of full length domains, demonstrating the importance of biological context, 4) it combines structural analysis with cell-based mutational and transcription activity analysis of the human ASCIZLC8 interaction that supports the model inferred from structural studies, and finally, 5) it pushes the envelope in the complexity and disorder of systems studied by both NMR and negative stain electron microscopy, and presents a model methodology for similarly complex systems.

[Editors' note: the author responses to the re-review follow.]

All three reviewers were impressed with the amount of new biophysical data added to the paper that considerably improved the characterization of this complex system. It was also felt that the literature was reviewed much more thoroughly this time and the current work framed in a better context. The consensus was that the biophysical characterization of a difficult system was strong, while the in-vitro and cellular experiments to prove the hypothesis regarding the feedback mechanism regulating expression were weaker. At issue is that while the overall trend regarding binding and activity is consistent with your claims, the effect is small (Figure 7D) and the errors are large, and in some cases it appears that the differences in the points corresponding to different Kd's are not statistically significant. It was the consensus of the reviewers that a revision should focus more on the biophysics, while suggesting biological relevance through the presentation of the transcription data. If possible, additional transcriptional experiments that attempt to decrease errors to strengthen this section would be recommended but are not required. Please provide a detailed point-by-point response to each of the issues raised below to assist the reviewers in an evaluation of your revised manuscript.1) As mentioned above the authors present an interesting and extensive biophysical study that characterizes the negative regulation of a transcription factor through binding of a main target protein. The reviewers feel that this (the biophysics) should be emphasized more strongly in a revised manuscript. It would strengthen the paper considerably if the errors associated with the transcription data of Figure 7 could be reduced, but this will be left at the discretion of the authors.

While we definitely recognize the value in reducing the error bars associated with our transcription activity assays, we do not think that repeating the assays would produce significantly better results because of the small changes observed between each sample. We have performed this assay many times and while the overall trend seen in Figure 7D is always consistent, the size of the error bars are as well.

In the revised manuscript, we have placed a greater emphasis on the biophysical characterization of the ASCIZ-LC8 interaction and the transcription data is presented to give weight and biological relevance to our hypothesis. Changes to the text can be seen in the last paragraph of the Introduction as well as the Discussion section.

2) In the third paragraph of the subsection “Structure and Distribution of LBD:LC8 Complexes Visualized by Single Particle EM” the authors state that high order oligomers are rare with fully formed complexes almost never observed. Yet in their model of Figure 8 they propose that under high concentrations of LC8 a fully formed complex is formed. There seems to be a discrepancy here.

It is true that high order (7+) oligomers are rare in comparison to low occupancy complexes. However, high order complexes of dLBD and hLBD bound to LC8 are formed in solution, as demonstrated by AUC (Figure 4), native gel (Figure 4), and negative stain electron microscopy (Figure 5, Figure 5—figure supplement 1). The language in the third paragraph of the subsection “Structure and Distribution of LBD:LC8 Complexes Visualized by Single Particle EM” has therefore been modified to state that these higher order oligomers are rare in comparison to the low occupancy complexes. While these high occupancy complexes are scarce, they are observed in every experiment.

Further it is hard for this reviewer to understand the role of negative cooperativity. If in fact the complete binding range is not used (i.e., one does not get fully bound transcription factor) then it seems like the full dynamic range over which changes in LC8 concentration occur is not optimally sampled by ASCIZ. What is the biological significance of the negative cooperativity, such that the full number of binding sites are not utilized? This seems not to be a very sensitive way of monitoring LC8 concentration.

This is an interesting point that highlights the unique aspects of the ASCIZ-LC8 interaction. The combination of positive and negative cooperativity that we observe leads to the formation of a dynamic mixture of low occupancy ASCIZ-LC8 complexes. The first few sites bind with positive cooperativity and binding thereafter becomes negatively cooperative, forcing the complex into a low occupancy state. These low occupancy complexes are a mixture of different stoichiometries that range from 2-4 for both dASCIZ and human ASCIZ.

While all 11 LC8 binding sites do not interact with LC8 in a linear fashion (i.e. occupancy does not increase linearly as LC8 concentration increases), the full number of binding sites are utilized. All sites interact with LC8, as demonstrated by NMR titration of dLBD with LC8 (Figure 6), but due to negative cooperativity, this interaction results in a heterogenous mixture of low occupancy complexes.

We propose that a population of low occupancy complexes is important to maintain a stable pool of cellular LC8. Instead of progressively binding more LC8 as it becomes available, our model suggests that ASCIZ’s multiple sites function as a “buffer” for LC8 concentration and aid in maintaining a homeostatic concentration. For example, an increase in LC8 cellular concentration would cause a small shift in the complex equilibrium towards a higher occupancy state and a corresponding decrease in the level of transcription, instead of causing a large change in both. Our transcription assays corroborate this idea, demonstrating a small change in transcriptional activity when the number of available binding sites changes (Figure 7). As LC8 is a ubiquitous protein that binds to hundreds of partners, the need to maintain a stable level of LC8 is likely necessary.